# Kernel Functional Optimisation

**Arun Kumar A V, Alistair Shilton, Santu Rana, Sunil Gupta, Svetha Venkatesh**
Applied Artificial Intelligence Institute ($A^2I^2$), Deakin University
Waurn Ponds, Geelong, Australia
`{aanjanapuravenk, alistair.shilton, santu.rana, sunil.gupta,`
`svetha.venkatesh}@deakin.edu.au`

## Abstract

Traditional methods for kernel selection rely on parametric kernel functions or a combination thereof and although the kernel hyperparameters are tuned, these methods often provide sub-optimal results due to the limitations induced by the parametric forms. In this paper, we propose a novel formulation for kernel selection using efficient Bayesian optimisation to find the best fitting non-parametric kernel. The kernel is expressed using a linear combination of functions sampled from a prior Gaussian Process (GP) defined by a hyperkernel. We also provide a mechanism to ensure the positive definiteness of the Gram matrix constructed using the resultant kernels. Our experimental results on GP regression and Support Vector Machine (SVM) classification tasks involving both synthetic functions and several real-world datasets show the superiority of our approach over the state-of-the-art.

## 1 Introduction

Kernel machines (Hofmann et al., 2008) generally work well with low-dimensional and small to medium-scaled data. In most kernel machines, the kernel function is chosen from the standard bag of popular kernels (Genton, 2001, Stein, 2015) such as Squared Exponential kernel (SE), Matérn kernel and Periodic kernel, or a weighted combination thereof (Aiolli and Donini, 2015, Gönen and Alpaydın, 2011, Rakotomamonjy et al., 2007). Recent developments (Jang et al., 2017, Wilson and Adams, 2013) in kernel learning parameterise the kernel function to boost the expressiveness of the kernel. However, the expressiveness of such kernels remains limited by the chosen parametric form and thus they often fall short in providing the best kernel function for complex data distributions.

There have been some early attempts to design an optimal non-parametric kernel to remove the limitations associated with the parametric forms. Ong et al. (2003, 2005) proposed a hyperkernel framework by defining a Reproducing Kernel Hilbert Space (RKHS) on the space of kernels *i.e.*, a kernel on kernels to support kernel learning. They formulate a semidefinite programming (Vandenberghe and Boyd, 1996) based optimisation problem using the representer theorem (Steinwart and Christmann, 2008, Vapnik, 1999) to find the best kernel. However, their method suffers from two key limitations: **(i)** their way of enforcing the positive definiteness property produces a restrictive search space, resulting in a sub-optimal solution, and **(ii)** the computational complexity of their method scales with the dataset size, making it infeasible for larger datasets. Benton et al. (2019) proposed Functional Kernel Learning (FKL), which extends the function space view of the Gaussian Process (GP) for kernel learning. FKL uses a transformed GP over a spectral density to define a distribution over kernels. However, the formulation of kernel functionals using the spectral densities induces strong assumptions on the properties such as periodicity, stationarity, *etc.* and thus are not generally applicable. Malkomes et al. (2016) proposed an automated kernel selection (BOMS) using Bayesian optimisation. The kernel space in BOMS is defined by the base kernels and the associated grammar to combine them. Although the search space is constructed by summing or multiplying the base kernels, the resultant kernel space is restricted in the compositional space of parametric forms.

35th Conference on Neural Information Processing Systems (NeurIPS 2021).

In this paper, we propose a generic framework called Kernel Functional Optimisation (KFO) to address the aforesaid shortcomings. First, it provides a flexible form of kernel learning whose computational complexity is decoupled from dataset size. Next, it allows us to use a computationally efficient Bayesian optimisation method to find the best kernel. We incorporate hyperkernels into our Bayesian framework that allows us to search for the optimal kernel in a Hilbert space of kernels spanned by the hyperkernel (Ong et al., 2005). We draw kernel functionals from a (hyper) GP distribution fitted using a hyperkernel. As the kernel drawn from the hyper-GP may be indefinite, we provide ways to ensure positive definiteness by transforming indefinite, or *Kreĭn* (Oglic and Gärtner, 2019, Ong et al., 2004) kernel space into a positive definite kernel space. The optimisation of kernel functionals necessitates solving larger covariance matrices and thus adds to the computational burden of the overall process. To speed up the computations, we perform a low-rank decomposition of the covariance matrix. Further, we provide a theoretical analysis of our method showing that it converges efficiently as in its cumulative regret grows only sub-linearly and eventually vanishes.

We evaluate the performance of our method on both synthetic and real-world datasets using SVM classification (Diehl and Cauwenberghs, 2003, Scholkopf and Smola, 2001, Burges, 1998) and GP regression tasks. Comparison of predictive performance against the state-of-the-art baselines demonstrates the superiority of our method. Further, we compare with the state-of-the-art performance reported in the latest survey paper on classifier comparison (Zhang et al., 2017) and find that our method provides the best performance on most of the datasets. Our main contributions in this paper are as follows: **(i)** we propose a novel approach for finding the best non-parametric kernel using hyperkernels and Bayesian functional optimisation (Section 3), **(ii)** we provide methods to ensure positive definiteness of the kernels optimised (Section 3), **(iii)** we derive the convergence guarantees to demonstrate that the regret grows sub-linearly for our proposed method (Section 4), **(iv)** we provide empirical results on both synthetic and real-world datasets to prove the usefulness (Section 5).

## 2   Background

**Notations**   We use lower case bold fonts $\mathbf{v}$ for vectors and $v_i$ for each element in $\mathbf{v}$. $\mathbf{v}^\intercal$ is the transpose. We use upper case bold fonts $\mathbf{M}$ (and bold greek symbols) for matrices and $M_{ij}$ for each element in $\mathbf{M}$. $|\cdot|$ for the absolute value. $\mathbb{N}_n = \{1, 2, \cdots, n\}$. $\mathbb{R}$ for Reals. $\mathcal{X}$ is a non-empty (index) set and $\mathbf{x} \in \mathcal{X}$. $\tilde{\mathcal{X}}$ is a non-empty (compounded index) set and $\tilde{\mathbf{x}} \in \tilde{\mathcal{X}}, \tilde{\mathcal{X}} = \mathcal{X}^2$. $(\cdot)_+$ clips a negative value to zero. $[\![\cdot]\!]$ is the Iverson bracket (Iverson, 1962) defined for any boolean value $I$ as $[\![I]\!] = 1$, if $I$ is True, 0 otherwise. Matrix $\mathbf{M} = [M_{ij}]_{i,j \in \mathbb{N}}$ and $\|\mathbf{M}\|_\text{F}$ is the Frobenius Norm of $\mathbf{M}$.

### 2.1   Bayesian Optimisation

Bayesian Optimisation (BO) (Brochu et al., 2010, Shahriari et al., 2015, Frazier, 2018) offers an elegant framework for finding the global extrema of an unknown, expensive and noisy function $f(\mathbf{x})$, represented as $\mathbf{x}^* = \operatorname{argmax}_{\mathbf{x} \in \mathcal{X}} f(\mathbf{x})$, where $\mathcal{X}$ is a compact search space. Bayesian optimisation is comprised of two main components: **(i)** a Gaussian Process (GP) (Williams and Rasmussen, 2006) model of $f$, and **(ii)** an acquisition function ($u$) (Kushner, 1964, Močkus, 1975, Wilson et al., 2018) to guide optimisation. Let $\mathcal{D} = \{\mathbf{x}_{1:t}, \mathbf{y}_{1:t}\}$ denote a set of observations of $f$, where $y = f(\mathbf{x}) + \epsilon'$ is the noisy observation corrupted with white Gaussian noise $\epsilon' \in \mathcal{N}(0, \sigma_{noise}^2)$. Then the predictive distribution at any point $\mathbf{x}_*$ is given as $f(\mathbf{x}_*)|\mathcal{D} \sim \mathcal{N}(\mu(\mathbf{x}_*), \sigma^2(\mathbf{x}_*))$, where $\mu(\mathbf{x}_*) = \mathbf{k}^\intercal[\mathbf{K} + \sigma_{noise}^2 \mathbf{I}]^{-1}\mathbf{y}_{1:t}$, $\sigma^2(\mathbf{x}_*) = k(\mathbf{x}_*, \mathbf{x}_*) - \mathbf{k}^\intercal[\mathbf{K} + \sigma_{noise}^2 \mathbf{I}]^{-1}\mathbf{k}$, $\mathbf{k} = [k(\mathbf{x}_*, \mathbf{x}_1) \cdots k(\mathbf{x}_*, \mathbf{x}_t)]$, $k : \mathcal{X} \times \mathcal{X} \to \mathbb{R}$ and $\mathbf{K} = [k(\mathbf{x}_i, \mathbf{x}_j)]_{i,j \in \mathbb{N}_t}$. The negative log-likelihood for a GP distribution is

$$-\log \mathcal{P}(y_*|\mathcal{D}, \mathbf{x}_*) = \tfrac{1}{2}\log(2\pi\sigma^2(\mathbf{x}_*)) + \tfrac{(y_* - \mu(\mathbf{x}_*))^2}{2\sigma^2(\mathbf{x}_*)} \tag{1}$$

The acquisition function ($u$) guides the search by balancing between exploitation (searching known high-value regions) and exploration (searching high-variance regions). Gaussian Process - Upper Confidence Bound (GP-UCB) acquisition function (Srinivas et al., 2012, Brochu et al., 2010) is the commonly used acquisition function to find the next best candidate for the evaluation, given as

$$u_t(\mathbf{x}) = \mu(\mathbf{x}) + \sqrt{\beta_t}\,\sigma(\mathbf{x}) \tag{2}$$

where $\beta_t$ grows as $O(\log t)$ with iteration $t$. Further, it can be shown that the average regret ($\mathcal{R} \triangleq \frac{1}{t}\sum_{t'=1}^{t} |f(\mathbf{x}^*) - f(\mathbf{x}_{t'})|$) grows as $O(\sqrt{\log t / t})$, and hence the average regret vanishes as $t \to \infty$. An algorithm for standard Bayesian optimisation is provided in the supplementary material.

The aforementioned standard Bayesian optimisation procedure often suffers from scaling issues originating from the curse of dimensionality. Wang et al. (2016) proposed REMBO - Random EMbedding Bayesian Optimisation - to address these scaling issues. REMBO works by projecting the objective function onto a lower-dimensional subspace prior to optimisation. LINEBO (Kirschner et al., 2019) builds on the same idea but instead of a fixed subspace, it decomposes the given black-box optimisation problem into a sequence of one-dimensional subproblems. Further, our method builds upon the principles of Bayesian functional optimisation methodologies (Vien et al., 2018, Vellanki et al., 2019, Shilton et al., 2020) in the literature to find a function to optimise the given process.

## 2.2 RKHS and Hyper-RKHS

The kernel functions used in the Gaussian process uniquely define an associated Reproducing Kernel Hilbert Space (RKHS) (Aronszajn, 1950). Formally:

**Definition 1:** *Let $\mathcal{H}_k$ be a Hilbert space of functions $f : \mathcal{X} \to \mathbb{R}$ on a non-empty set $\mathcal{X}$. A function $k : \mathcal{X} \times \mathcal{X} \to \mathbb{R}$ is a reproducing kernel of $\mathcal{H}_k$, and $\mathcal{H}_k$ a* **Reproducing Kernel Hilbert Space (RKHS)**, *if the following properties are satisfied.*

- $k$ spans $\mathcal{H}_k$ i.e., $\mathcal{H}_k = \overline{\mathrm{span}\{k(\cdot, \mathbf{x}) | \mathbf{x} \in \mathcal{X}\}}$
- $\forall \mathbf{x} \in \mathcal{X},\ \forall f \in \mathcal{H}_k,\ \langle f(\cdot), k(\cdot, \mathbf{x}) \rangle_{\mathcal{H}_k} = f(\mathbf{x})$ *(the reproducing property)*
- $\forall \mathbf{x},\ \mathbf{x}' \in \mathcal{X},\ k(\mathbf{x}, \mathbf{x}') = \langle k(\cdot, \mathbf{x}), k(\cdot, \mathbf{x}') \rangle_{\mathcal{H}_k}$

Next, we consider the Reproducing Kernel Hilbert Space (RKHS) of kernels by introducing a compounded index set $\tilde{\mathcal{X}} : \mathcal{X} \times \mathcal{X}$ and a hyperkernel $\kappa$ (Ong and Smola, 2003, Ong et al., 2003). Analogous to the RKHS (Aronszajn, 1950) associated with the kernel function, a hyperkernel defines an associated Hyper-Reproducing Kernel Hilbert Space (Hyper-RKHS) (Ong et al., 2003).

**Definition 2:** *Let $\mathcal{X}$ be a non-empty set and $\tilde{\mathcal{X}}$ denote $\mathcal{X} \times \mathcal{X}$. The Hilbert space $\mathcal{H}_\kappa$ of functions $k : \tilde{\mathcal{X}} \to \mathbb{R}$ is called a* **Hyper-Reproducing Kernel Hilbert Space (Hyper-RKHS),** *if there exists a hyperkernel $\kappa : \tilde{\mathcal{X}} \times \tilde{\mathcal{X}} \to \mathbb{R}$ that satisfies the following properties:*

- $\kappa$ spans $\mathcal{H}_\kappa$ i.e., $\mathcal{H}_\kappa = \overline{\mathrm{span}\{\kappa(\cdot, \tilde{\mathbf{x}}) \mid \tilde{\mathbf{x}} \in \tilde{\mathcal{X}}\}}$
- $\forall \tilde{\mathbf{x}} \in \tilde{\mathcal{X}},\ \forall k \in \mathcal{H}_\kappa,\ \langle k(\cdot), \kappa(\cdot, \tilde{\mathbf{x}}) \rangle_{\mathcal{H}_\kappa} = k(\tilde{\mathbf{x}})$ *(the reproducing property)*
- $\forall \tilde{\mathbf{x}},\ \tilde{\mathbf{x}}' \in \tilde{\mathcal{X}},\ \kappa(\tilde{\mathbf{x}}, \tilde{\mathbf{x}}') = \langle \kappa(\cdot, \tilde{\mathbf{x}}), \kappa(\cdot, \tilde{\mathbf{x}}') \rangle_{\mathcal{H}_\kappa}$
- $\kappa(\mathbf{x}', \mathbf{x}'', \mathbf{x}''', \mathbf{x}'''') = \kappa(\mathbf{x}'', \mathbf{x}', \mathbf{x}''', \mathbf{x}'''')\ \forall \mathbf{x}', \mathbf{x}'', \mathbf{x}''', \mathbf{x}'''' \in \mathcal{X}$

The GP distribution defined by a hyperkernel $\kappa$ is a distribution on the space of kernels. This Hyper-RKHS is a Hilbert space comprised of positive definite, negative definite and indefinite kernels. A *Kreĭn* kernel $k$ (Oglic and Gärtner, 2018, Ong et al., 2004) is an indefinite kernel with a positive decomposition *i.e.*, there exist positive kernels $k_+ \in \mathcal{H}_+$ and $k_- \in \mathcal{H}_-$, such that $k = k_+ - k_-$. From Definition 2, we see that $\kappa(\tilde{\mathbf{x}}, \tilde{\mathbf{x}}') = \kappa(\mathbf{x}', \mathbf{x}'', \mathbf{x}''', \mathbf{x}'''')$ is a kernel, where $\tilde{\mathbf{x}} = (\mathbf{x}', \mathbf{x}'')$. Generally, the samples drawn from $\mathcal{GP}(0, k)$ do not lie in the corresponding RKHS $\mathcal{H}_k$, but in a larger RKHS $\mathcal{H}_{k' \neq k}$ (see discussion in Kanagawa et al. (2018), Remark 3.8 and Section 4). We also note that the posterior mean of $\mathcal{GP}(0, k)$ lies in the RKHS $\mathcal{H}_k$. Similarly, with hyper-GP, the samples drawn from $\mathcal{GP}_\kappa(0, \kappa)$ lie in RKHS $\mathcal{H}_{\kappa' \neq \kappa}$, whereas its posterior mean ($\mu$) lies in $\mathcal{H}_\kappa$. Further, $\mu$ can be decomposed with positive and negative weights as $\mu = \mu_+ - \mu_- = \sum_i \alpha_{i_+} \kappa(\cdot, \tilde{\mathbf{x}}_{i_+}) - \sum_i \alpha_{i_-} \kappa(\cdot, \tilde{\mathbf{x}}_{i_-})$, where $\alpha_{i_+}, \alpha_{i_-} > 0$; and $\mu_\pm = \sum_i \alpha_{i_\pm} \kappa(\cdot, \tilde{\mathbf{x}}_{i_\pm})$ is a kernel (Definition 2 and Ong et al. (2004)). Thus, $\mu = \mu_+ - \mu_-$ is a *Kreĭn* kernel (Oglic and Gärtner, 2019).

## 3 Framework

In this paper, we address the global optimisation problem formulated as $K^* = \mathrm{argmax}_{K \in \mathcal{H}_\kappa} f(K)$, where $f : \mathcal{H}_\kappa \to \mathbb{R}$ is an expensive objective functional and $\kappa$ is a hyperkernel. In particular, we are interested in finding the best kernel $K^* \in \mathcal{H}_\kappa$ to maximise the model performance represented by the objective functional $f$ (for example, $f$ can be the leave-one-out classification performance of a SVM classifier). First, we describe the construction of valid kernel functionals using hyperkernel, followed by a discussion on the kernel functional optimisation using Bayesian optimisation. A flowchart

describing the overall optimisation process of kernel functionals is shown in Figure 1. A complete algorithm for the Kernel Functional Optimisation (KFO) is given by Algorithm 1.

## 3.1 Construction of Kernel Functionals from Hyper-Gaussian Process

Ong and Smola (2003) and Ong et al. (2003, 2005) have discussed the general guidelines to design a hyperkernel. We follow the same strategy to formulate *Matérn Harmonic Hyperkernel* ($\kappa$):

$$\kappa(\mathbf{x}, \mathbf{x}', \mathbf{x}'', \mathbf{x}''') = \frac{1 - \lambda_h}{1 - \left(\lambda_h \, \bar{c}_1 \, \bar{c}_2 \, \exp\left(-\frac{\sqrt{3}}{l}(r_1 + r_2)\right)\right)} \tag{3}$$

where $\lambda_h$ and $l$ correspond to the hyperparameters of the hyperkernel, $r_1 = \|\mathbf{x} - \mathbf{x}'\|$, $r_2 = \|\mathbf{x}'' - \mathbf{x}'''\|$, $\bar{c}_1 = \left(1 + \frac{\sqrt{3}}{l} r_1\right)$, and $\bar{c}_2 = \left(1 + \frac{\sqrt{3}}{l} r_2\right)$. The derivation of Matérn Harmonic Hyperkernel is provided in the supplementary material. In our proposed method, we use the draws from a (hyper) Gaussian process $\mathcal{GP}_\kappa(0, \kappa)$ to construct finite-dimensional subspaces of our kernel space on which we perform optimisation. As discussed in Section 2.2, the kernel samples drawn from $\mathcal{GP}_\kappa(0, \kappa)$ do not lie in $\mathcal{H}_\kappa$, hence we approximate the draws using the posterior mean of $\mathcal{GP}_\kappa(0, \kappa)$ lying in $\mathcal{H}_\kappa$.

In practice, when sampling from $\mathcal{GP}_\kappa(0, \kappa)$, we assume a grid $\mathcal{G}$ with $N_g$ points $\{\tilde{\mathbf{x}}_1, \tilde{\mathbf{x}}_2, \cdots | \tilde{\mathbf{x}}_i \in \tilde{\mathcal{X}} : \mathcal{X} \times \mathcal{X}, \forall i \in \mathbb{N}_{N_g}\}$ for placing a GP distribution on kernels using a hyperkernel $\kappa$ mentioned in Eq. (3). The sample set $\mathbf{k} \sim \mathcal{GP}_\kappa(0, \kappa)$ is essentially a set of noiseless observations of the kernel $K$ on the grid-points $\tilde{\mathbf{x}}_1, \tilde{\mathbf{x}}_2, \cdots$ lying in $\mathcal{H}_{\kappa' \neq \kappa}$. The number of points in the grid is chosen such that the resulting grid is sufficiently fine to represent the kernel $K$ everywhere on $\tilde{\mathcal{X}}$. Therefore, for any point $\tilde{\mathbf{x}}_i \in \tilde{\mathcal{X}}$, the posterior variance of the kernel $K$ given the observations $\{(\tilde{\mathbf{x}}_i, k_i) | i \in \mathbb{N}_{N_g}\}$ is negligible and thus the kernel $K$ can be approximated using the posterior mean of $\mathcal{GP}_\kappa(0, \kappa)$ as

$$K(\tilde{\mathbf{x}}) \approx [\kappa(\tilde{\mathbf{x}}, \tilde{\mathbf{x}}_1) \, \kappa(\tilde{\mathbf{x}}, \tilde{\mathbf{x}}_2) \, \kappa(\tilde{\mathbf{x}}, \tilde{\mathbf{x}}_3) \cdots] \, \boldsymbol{\kappa}^{-1} \, \mathbf{k} = \sum_i \alpha_i \, \kappa(\tilde{\mathbf{x}}, \tilde{\mathbf{x}}_i), \text{ where } \boldsymbol{\alpha} = \boldsymbol{\kappa}^{-1} \, \mathbf{k} \tag{4}$$

A very fine resolution grid ensures that we can capture small-scale patterns in the kernel. However, a large grid size comes with large computational costs. Therefore, the choice of $N_g$ is a trade-off between the overall computational cost and the accuracy of kernel optimisation expected. We discuss the computational complexity and the associated memory demands pertaining to $N_g$ in Section 4.4.

## 3.2 Kernel Functional Optimisation

We adopt the ideas from Bayesian optimisation method - LINEBO (Kirschner et al., 2019) for the optimisation of non-parametric kernel functionals via a sequence of one-dimensional projections. First, we discuss the construction of low-dimensional subspaces. The key challenge here is to address the computational burden with the use of large grid. Next, we describe the Bayesian functional optimisation for each of the subspace and across many such subspaces. Since the best kernel obtained is a *Kreĭn* kernel, we apply transformations to ensure the positive definiteness of the Gram matrix.

**Construction of Low-dimensional Spaces** We start with the construction of low-dimensional search space spanned by randomly chosen basis vectors drawn from the hyper-GP $\mathcal{GP}_\kappa(0, \kappa)$. The hyper-GP surrogate modelling requires the computation of covariance matrix $\boldsymbol{\kappa} \in \mathbb{R}^{N_g \times N_g}$ using $\kappa$ for the predefined grid $\mathcal{G}$. Further, the accuracy of the kernel functional to represent the kernel $K$ is directly proportional to the assumed grid size $N_g$. To avoid the computational burden arising

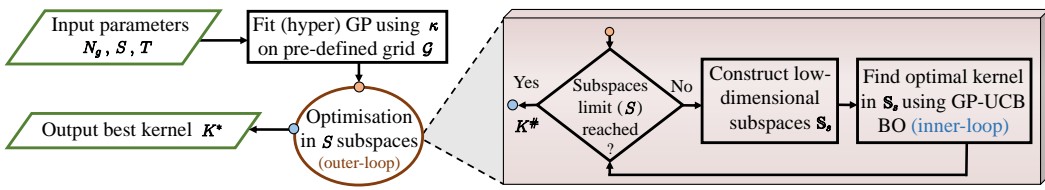

Figure 1: A complete flowchart for the Kernel Functional Optimisation (KFO) framework.

from the larger grid size $N_g$, we perform Principal Component Analysis (PCA) (Wold et al., 1987) and choose $N'$ principal components. Mathematically, we represent $\boldsymbol{\kappa} = (\boldsymbol{E}\sqrt{\boldsymbol{\Lambda}})(\boldsymbol{E}\sqrt{\boldsymbol{\Lambda}})^{\mathsf{T}}$, where $i^{th}$ column $\boldsymbol{e}_i$ in $\boldsymbol{E} \in \mathbb{R}^{N_g \times N'}$ corresponds to the $i^{th}$ principal component and $\boldsymbol{\Lambda} \in \mathbb{R}^{N' \times N'}$ is the diagonal matrix containing top $N'$ eigenvalues. The outer-loop in Algorithm 1 iterates through a sequence of $S$ $d$-dimensional subspaces by drawing $d$ random basis vectors in each subspace from $\mathcal{GP}_\kappa(0, \kappa)$ *i.e.*, $\mathbf{k}^{(1)}, \mathbf{k}^{(2)}, \cdots, \mathbf{k}^{(d)} \sim \mathcal{GP}_\kappa(0, \kappa)$, where $\mathbf{k}^{(\cdot)} = \boldsymbol{E}\sqrt{\boldsymbol{\Lambda}} \cdot \boldsymbol{\beta}^{(\cdot)}$ and $\boldsymbol{\beta}^{(\cdot)} \sim \mathcal{N}(0, \mathbf{I}_{N'})$.

**Kernel Optimisation Observation Model**  As discussed earlier, we construct kernel functionals $K(\cdot, \cdot)$ from the hyper-GP distribution $\mathcal{GP}_\kappa(0, \kappa)$ as per Eq. (4) using

$$\mathbf{k} = K^{\#} + \lambda^{(1)}\mathbf{k}^{(1)} + \cdots + \lambda^{(d)}\mathbf{k}^{(d)} \tag{5}$$

where $\lambda^{(\cdot)} \in [0, 1]$, $\mathbf{k}^{(\cdot)}$ are the random basis vectors drawn and $K^{\#}$ corresponds to the best kernel found across all the previous subspaces. The optimal kernel in the given subspace $s$ is obtained by optimising $\boldsymbol{\lambda}$ using a Bayesian optimisation procedure with another GP distribution $\mathcal{GP}(0, \overline{k}_{\text{SE}})$. The observation model for $\mathcal{GP}(0, \overline{k}_{\text{SE}})$ is considered as $\mathcal{D}'_s = \{(K, y = f(K))\}$, where $K$ is the kernel functional constructed and $y$ is a measure signifying the ability of the latent kernel to represent the given data. For example, log-likelihood can be used as the measure $y$ in our observation model.

**Building GP for Kernel Optimisation**  We fit a GP distribution $\mathcal{GP}(0, \overline{k}_{\text{SE}})$ on the observed kernel functionals using the Squared Exponential (SE) kernel ($\overline{k}_{\text{SE}}$) given by

$$\overline{k}_{\text{SE}}(K_1, K_2) = \overline{\sigma}_f^2 \exp\left( \frac{-1}{2\overline{\Upsilon}^2} \left\| K_1 - K_2 \right\|_{\mathcal{H}_{\kappa' \neq \kappa}}^2 \right) \tag{6}$$

where $\overline{\sigma}_f^2$ and $\overline{\Upsilon}$ correspond to the signal variance and lengthscale parameters of $\overline{k}_{\text{SE}}$. Although there is no restriction on the kernel choice here, we consider the commonly used SE kernel. As mentioned earlier, we approximate $K$ using the posterior mean ($\mu$), therefore we compute the similarity between kernel functionals using the RKHS norm ($\| \cdot \|_{\mathcal{H}_\kappa}$) estimated as

$$\|K_1 - K_2\|_{\mathcal{H}_{\kappa' \neq \kappa}} \approx \|\mu_1 - \mu_2\|_{\mathcal{H}_\kappa} = \sqrt{\boldsymbol{\alpha}_1^{\mathsf{T}}\boldsymbol{\kappa}\boldsymbol{\alpha}_1 + \boldsymbol{\alpha}_2^{\mathsf{T}}\boldsymbol{\kappa}\boldsymbol{\alpha}_2 - 2\boldsymbol{\alpha}_1^{\mathsf{T}}\boldsymbol{\kappa}\boldsymbol{\alpha}_2} \tag{7}$$

where $\mu_1$ and $\mu_2$ are the posterior mean approximations of $K_1$ and $K_2$, respectively. We refer to the supplementary material for the details of similarity formulations using $L_2-$Norm.

**Kernel Optimisation**  We find the best kernel functional in the given low-dimensional subspace using GP-UCB acquisition function (Eq. (2)) with $\beta_t = 2\log(t^{2+\frac{\tilde{n}}{2}}\pi^2/3\tilde{\delta})$, where $\tilde{n}$ corresponds to the total number of kernel functional observations and $\tilde{\delta}$ is a value in $[0, 1]$. The best kernel found ($K^{\#}$) across all the previous subspaces acts as a subspace bias guiding the optimisation in the subsequent subspaces as per Eq. (5). The selection of $S$ $d$-dimensional subspaces (outer-loop) and optimising the kernel (for $T$ iterations) in each of the subspace (inner-loop) continues until the search budget is exhausted. The hyperparameters $\overline{\theta} = \{\overline{\sigma}_f^2, \overline{\Upsilon}\}$ in $\overline{k}_{\text{SE}}$ are tuned by maximising the log marginal likelihood. In addition to that, the hyperparameters of the hyperkernel ($\Theta = \{\lambda_h, l\}$) mentioned in Eq. (3) are tuned using another standard Bayesian optimisation procedure. The observation model for the hyperparameter tuning of hyperkernel is constructed as $\mathfrak{D} = \{(\Theta, y' = \Gamma(\Theta))\}$, where $\Gamma$ maps the model performance $y'$ with the corresponding hyperparameter set $\Theta$. We refer to the supplementary material for the detailed discussion on tuning the hyperparameters of both kernel and hyperkernel.

**From *Kreĭn* kernels to Positive Definite Gram Matrix**

As the kernel approximated by Eq. (4) is an indefinite, or *Kreĭn* kernel ($K$), the Gram matrix ($\mathbf{C}$) constructed for the datapoints using $K$ is also indefinite. We use the following matrix post-processing methods to ensure the positive definiteness of the Gram matrix constructed.

The Eigen Value Decomposition (EVD) based matrix post-processing involves the decomposition of the Gram matrix $\mathbf{C}$ as $\mathbf{C} = \mathbf{Z}\boldsymbol{\Delta}\mathbf{Z}^{\mathsf{T}}$, where $\mathbf{Z}$ is the square matrix containing eigenvectors corresponding to the eigenvalues in the diagonal matrix $\boldsymbol{\Delta}$. The Eigen spectrum clip ($\Delta_{ii} = (\Delta_{ii})_+$) ensures positive definiteness of the given training and test covariance matrix, but in isolation, without considering the transformation of the underlying kernel function, thus resulting in inconsistency

---

**Algorithm 1** Kernel Functional Optimisation

---

**Input**: $N_g$ - Number of points in the grid, $S$ - Number of subspaces search, $T$ - Number of iterations

1. Initialise $(K^{\#}, y_{\text{best}}) \leftarrow (\mathbf{0}, 0)$, $\mathcal{D}_0 \leftarrow \emptyset$
2. Compute $\boldsymbol{\kappa}$ for $N_g$ grid points $\tilde{\mathbf{x}}_1, \tilde{\mathbf{x}}_2, \cdots$ using Eq. (3)
3. Perform PCA of $\boldsymbol{\kappa}$ as $\boldsymbol{\kappa} = (\boldsymbol{E}\sqrt{\boldsymbol{\Lambda}})(\boldsymbol{E}\sqrt{\boldsymbol{\Lambda}})^{\mathsf{T}}$
4. **for** Subspace $s = 1, 2, \cdots, S$ **do** (outer-loop)
5. $\quad$ Sample $\mathbf{k}^{(1)}, \mathbf{k}^{(2)}, \cdots, \mathbf{k}^{(d)} \sim \mathcal{GP}_{\kappa}(0, \kappa)$
6. $\quad$ Generate random initial observations in the current subspace $s$
   $$\mathcal{D}'_s = \{(K, y) \mid K \xleftarrow{\text{Eq. (4)}} K^{\#} + \lambda^{(1)}\mathbf{k}^{(1)} + \cdots + \lambda^{(d)}\mathbf{k}^{(d)}, y = f(K), \lambda_{i \in \mathbb{N}_d} \sim U(0, 1)\}$$
7. $\quad$ **for** each iteration $t = 1, 2, \cdots, T$ **do** (inner-loop)
8. $\quad\quad$ Solve $\boldsymbol{\lambda}_* = \underset{\boldsymbol{\lambda} \in [0,1]^d}{\arg\max} \; u_t(\mu(K(\boldsymbol{\lambda})) + \sqrt{\beta_t}\,\sigma(K(\boldsymbol{\lambda})))$
9. $\quad\quad$ Compute the new kernel $K_{\text{new}}$ as $K_{\text{new}} \xleftarrow{\text{Eq. (4)}} K^{\#} + \lambda_*^{(1)}\mathbf{k}^{(1)} + \cdots + \lambda_*^{(d)}\mathbf{k}^{(d)}$
10. $\quad\quad$ Use the kernel $K_{\text{new}}$ and $\widehat{\mathbf{C}}$ to measure the fitting quality $y$ as $y_{\text{new}} = f(K_{\text{new}})$
11. $\quad\quad$ $\mathcal{D}'_s \leftarrow \mathcal{D}'_s \cup \{(K_{\text{new}}, y_{\text{new}})\}$
12. $\quad$ **end for**
13. $\quad$ $\mathcal{D}_s \leftarrow \mathcal{D}_{s-1} \cup \mathcal{D}'_s$
14. $\quad$ $(K^{\#}, y_{\text{best}}) = \underset{(K,y) \in \mathcal{D}_s}{\arg\max} \; y$
15. **end for**
16. $K^* \leftarrow K^{\#}$
17. return $(K^*, y_{\text{best}})$

---

(see discussion 2.2 in Chen et al. (2009)). Therefore, to consistently transform both the training and test points, the Eigen spectrum clip is treated as a linear transformation on the training points first *i.e.,* $\widehat{\mathbf{C}}_{\text{train}} = \boldsymbol{\vartheta}_{\text{clip}}\mathbf{C}_{\text{train}}$, where $\boldsymbol{\vartheta}_{\text{clip}}$ is the spectrum transformation matrix and then, apply the same transformation on $\mathbf{c}_{\text{test}} = [K(\mathbf{x}_{\text{test}}, \mathbf{x}_1) K(\mathbf{x}_{\text{test}}, \mathbf{x}_2) \cdots]^{\mathsf{T}}$ as $\widehat{\mathbf{c}}_{\text{test}} = \boldsymbol{\vartheta}_{\text{clip}}\mathbf{c}_{\text{test}}$, where $\boldsymbol{\vartheta}_{\text{clip}} = \mathbf{Z}\boldsymbol{\Delta}_{\text{clip}}\mathbf{Z}^{\mathsf{T}}$ and $\boldsymbol{\Delta}_{\text{clip}} = \text{diag}(\llbracket\Delta_{11} \geq 0\rrbracket, \llbracket\Delta_{22} \geq 0\rrbracket, \cdots)$. The magnitude of change in the transformed matrix $(\widehat{\mathbf{C}})$ from the given indefinite kernel matrix $(\mathbf{C})$ is minimum with the spectrum clip transformations *i.e.,* $\widehat{\mathbf{C}}_{\text{clip}} = \arg\min_{\widehat{\mathbf{C}} \succcurlyeq 0} \|\mathbf{C} - \widehat{\mathbf{C}}\|_{\text{F}}$. We note that, it is possible to use the original optimised kernel for specialised SVMs (Ying et al., 2009), but we consider this as part of the future work.

For GPs, there is a strong requirement that the covariance matrix is positive definite as it needs to generate positive definite covariances. Ayhan and Chu (2012) have demonstrated the vulnerabilities of GP with indefinite kernels. The aforestated EVD based post-processing gets complicated for GP. The GP predictive distribution involves the calculation of mean $\mu(\cdot)$ and variance $\sigma^2(\cdot)$ for the test samples. The variance requires the computation of $[K(\mathbf{x}_{\text{test}}, \mathbf{x}_{\text{test}})]$. Although the linear transformation $\boldsymbol{\vartheta}_{\text{clip}}$ on $\mathbf{C}_{\text{train}}$ ensures positive definiteness of $\mathbf{c}_{\text{test}} = [K(\mathbf{x}_{\text{test}}, \mathbf{x}_1) K(\mathbf{x}_{\text{test}}, \mathbf{x}_2) \cdots]^{\mathsf{T}}$, it does not consistently transform $[K(\mathbf{x}_{\text{test}}, \mathbf{x}_{\text{test}})]$. Therefore, we need ways to enforce positive definiteness before we compute predictive variances. To ensure positive definiteness in GPs, we clip the values of $\boldsymbol{\alpha}$ *i.e.,* $\boldsymbol{\alpha} = [(\alpha_i)_+]$ in the posterior mean approximation of kernels by visualising the kernel approximation (Eq. (4)) in terms of the representer theory mentioned in Ong et al. (2005).

## 4 Theoretical Analysis

### 4.1 Inner-loop

The cumulative regret for the optimisation in the inner-loop is given as $R_T = \sum_{t=1}^{T} f(K^*) - f(K_t)$, where $K^*$ is the best kernel found across all the subspaces. In the inner-loop, our goal is to derive the upper bound for the cumulative regret $(R_T)$ in terms of the total number of iterations $T$.

In conventional BO algorithms, the variables being optimised are directly used in the model construction. In contrast, the inner-loop in our proposed method constructs the model using the projection of the variables $(\boldsymbol{\lambda}_*)$ being optimised in the functional space *i.e.,* $\mathbf{k} = K^{\#} + \sum_i \lambda^{(i)}\mathbf{k}^{(i)}$.

**Proposition 1**: *Let $\mathbb{S}_s$ be the subspace constructed in each instance $s$ of the outer-loop. Then, at each iteration $t$ of the inner-loop, the maximum information gain ($\gamma_t$) of the kernel $\overline{k} : \mathbb{S}_s \times \mathbb{S}_s \to \mathbb{R}$ is same as that of the information gain of the standard kernel in Euclidean space $k : \mathcal{X} \times \mathcal{X} \to \mathbb{R}$.*

The proof of proposition 1 is deferred to the supplementary material.

It is important to note that the model for $f$ in the inner-loop is constructed with the observations obtained from the current and previous subspaces search and not just the observations from the current search. Therefore, the bounds on the overall regret for the inner-loop can be derived as follows.

**Theorem 1**: *Let $f(K)|\mathcal{D}_{s-1}$ be the posterior of $f$ in subspace $s$ before entering the inner-loop and $f(K)|\mathcal{D}_{s-1} \cup \mathcal{D}'_s$ be the posterior at iteration $t$ of the inner-loop. Then, the updated posterior $f(K)|\mathcal{D}_{s-1} \cup \mathcal{D}'_s$ is equivalent to the posterior of the biased GP with prior covariance $\widehat{k}_{\mathcal{D}_{s-1}}$ and the inner-loop regret grows sub-linearly as $\mathcal{O}^*(\sqrt{dt\gamma_{\mathcal{D}_{s-1},t}})$, where $\gamma_{\mathcal{D}_{s-1},t}$ is the maximum information gain for the prior covariance $\widehat{k}_{\mathcal{D}_{s-1}}$ and $\mathcal{O}^*$ notation is a variation of $\mathcal{O}$ with log factors suppressed.*

The proof of Theorem 1 is provided in the supplementary material.

## 4.2 Outer-loop

We provide a theoretical analysis of the outer-loop based on the notion of *effective dimension* (Kirschner et al., 2019, Wang et al., 2016). As we deal with the functionals in our proposed method, the standard definition of *effective dimension* is slightly modified as follows:

**Definition 3**: *A function $f : \mathcal{H}_\kappa \to \mathbb{R}$ is said to have effective dimensionality $d' \in \mathbb{N}$, if there exists $\mathbf{k}^{(1)}, \mathbf{k}^{(2)}, \cdots, \mathbf{k}^{(d')} \in \mathcal{H}_\kappa$, such that $\|f(K + K_\perp) - f(K)\| = 0, \forall K \in \mathbb{K}, \forall K_\perp \in \mathbb{K}^\perp$, where $\mathbb{K} = \mathrm{span}(\mathbf{k}^{(1)}, \mathbf{k}^{(2)}, \cdots, \mathbf{k}^{(d')})$ and $\mathbb{K}^\perp = \{\tilde{K} \in \mathcal{H}_\kappa \mid \langle K, \tilde{K} \rangle_{\mathcal{H}_\kappa} = 0, \forall K \in \mathbb{K}\}$.*

Following Kirschner et al. (2019), we derive the regret bounds for the outer-loop.

**Theorem 2:** *Given a twice Frechet-differentiable kernel $\overline{k} : \mathcal{H}_\kappa \times \mathcal{H}_\kappa \to \mathbb{R}$, let $0 < \delta < 1$, $f \sim \mathcal{GP}(0, k)$ with effective dimension $d'$ and maxima $K^* = \mathrm{argmax}_{K \in \mathcal{H}_\kappa} f(K)$. Then, after $s$ subspaces search ($s$ outer-loop iterations), with probability at least $1 - \delta$, the regret $f(K^*) - f(K^\#) \in \mathcal{O}(\llbracket d < d' \rrbracket (\frac{1}{s} \log(\frac{1}{\delta}))^{\frac{2}{d'-d}} + \epsilon_{d,\delta})$, where $K^\#$ is the best kernel found across all the previous subspace searches and $\epsilon_{d,\delta}$ is the regret bound for the inner-loop and $\llbracket \cdot \rrbracket$ is the Iverson bracket.*

The proof of Theorem 2 is provided in the supplementary material.

## 4.3 Overall Convergence

In LINEBO, one-dimensional subspaces (or the lines) are optimised up to $\mathtt{err}(K^+) < \epsilon$ for some fixed $\epsilon$ (Lemma 4 of Kirschner et al. (2019)) and $K^+ = \mathrm{argmax}_{K_i \in K_{1:t}} f(K_i)$. In our method, for a given subspace $s$, we terminate after $T$ iterations with accuracy $\mathtt{err}(K^+) \leq \epsilon_{d,\delta}$. In our setup with $d = 1$, given a fixed budget ($T$ iterations) for the inner-loop, we get $\epsilon_{1,\delta} \in \mathcal{O}(T^{c-\frac{1}{2}})$, where $c \in (0, 0.5)$ (Assumption 2 in Kirschner et al. (2019)). On the other hand, if the number of vectors ($d$) spanning the random basis is same as the effective dimensionality ($d'$), then our convergence is analogous to REMBO (Wang et al., 2016), with the regret imposed only by $\epsilon_{d',\delta}$. Further, the order of regret bound in such cases remains unchanged even if we consider only one subspace search ($S = 1$).

Alternatively, simple regret measure implemented as a terminating condition in the inner-loop results in the regret bound $\epsilon_{d,\delta} = \epsilon$. If we consider one-dimensional spaces ($d = 1$) and use $\mathtt{err}(K^+) < \epsilon$ as the terminating condition for the inner-loop, the convergence guarantee of our algorithm is exactly same as that of LINEBO with $\epsilon_{d,\delta} = \epsilon$. Thus, the inner-loop of our algorithm is expected to complete in $T \in \mathcal{O}(\epsilon^{\frac{2}{1-2c}})$ iterations for some $c \in (0, 0.5)$ (see discussion around Assumption 2 in Kirschner et al. (2019)), resulting in $\mathcal{O}(S\epsilon^{\frac{2}{1-2c}})$ total number of function evaluations overall.

## 4.4 Computational Analysis

The computational complexity of our approach is in the order of $\mathcal{O}(STN_g^3)$, where $S$ is the number of subspace searches, $T$ is the number of iterations in each subspace and $N_g$ is the number of points in the grid, without including the complexity of the downstream class (as it would be different for

different kernel machines). The main bottleneck of our method is the computation of the covariance matrix $\boldsymbol{\kappa} \in \mathbb{R}^{N_g \times N_g}$. To avoid the computational burden resulting from the large covariance matrix $\boldsymbol{\kappa}$ for the given $N_g$, we perform Principal Component Analysis (PCA) of $\boldsymbol{\kappa}$. Here, we do not perform a full PCA, rather we choose only top $N'$ principal components ($N' \ll N_g$). The computational complexity of finding top $N'$ principal components is $\mathcal{O}(N' N_g^2)$, which is much lower than $\mathcal{O}(N_g^3)$. Moreover, we perform PCA only once, prior to entering the outer and inner optimisation loops. Thus, we incur a cost on startup but are rewarded with significant computational savings in the main optimisation loop where the computational burden is proportional to $N'$ rather than $N_g^2$. The memory complexity for optimising the kernel functionals using our proposed method is in the order of $\mathcal{O}(N_g^2)$.

Further, as we deal with a kernel selection problem, we are only concerned with the complexity of the observed search (kernel) space. Theoretically, the optimality of our method is not limited to any dataset-specific characteristics such as the number of dimensions ($n$) or the number of target classes in the given problem. Such characteristics do not have a significant role in the kernel optimisation, but the complexity of the given search (kernel) space plays a vital role in the optimisation performance.

## 5    Experiments

We evaluate the performance of our proposed algorithm (KFO) on synthetic benchmark functions and also apply our method on real-world datasets for SVM classification and GP regression tasks. We have considered the following experimental settings for KFO. We have used Matérn Harmonic Hyperkernel (Eq. (3)) to define the space of kernel functionals. To express the kernel as kernel functional in Hyper-RKHS, we consider $N_g \gtrsim 10 \times n$ for a given $n$ dimensional problem. The outer-loop representing the number of low-dimensional subspace searches ($S$) to find the best kernel function is restricted to $S = 5$ and the number of iterations ($T$) in each of the subspace (inner-loop) is restricted to $T = 20$. We use GP-UCB acquisition function to guide the search for optimum in all our experiments and at all levels. The hyperparameters $\lambda_h$ and $l$ of the hyperkernel (Eq. (3)) are tuned in the interval $(0, 1]$ using a standard BO procedure mentioned in the supplementary material.

### 5.1    Synthetic Experiments

In this experiment, we test our algorithm (KFO) with the following synthetic functions: **(i)** Triangular wave, **(ii)** a mixture of three Gaussian distributions ($G_{\mathrm{mix}}$), and **(iii)** SINC function. We compare with the following stationary and non-stationary kernels: **(i)** SE kernel, **(ii)** Matérn kernel with $\nu = 3/2$ (Mat3/2), and **(iii)** Multi-Kernel Learning (MKL) as a linear combination of SE, Mat3/2 and Linear kernel. The hyperparameters $\Upsilon$, $\sigma_f^2$ and weights $\mathbf{w}$ (in the case of MKL) of the baseline kernels are tuned by maximising the log-likelihood. We compute the posterior distributions for the aforesaid synthetic functions. We report the mean and the standard deviation of the maximum log-likelihood computed over 10 random runs. We show the posterior distribution and the maximum log-likelihood estimates obtained for Triangular wave function in Figure 2. We refer to the supplementary material for the results on other synthetic functions. It is evident that the posterior distribution computed using the standard kernels has poor predictions in the held-out test region. By contrast, the kernel suggested by KFO has better predictive mean and variance in the held-out test region. Especially note that the KFO optimised kernel was able to find the correct periodicity even without explicit enforcement.

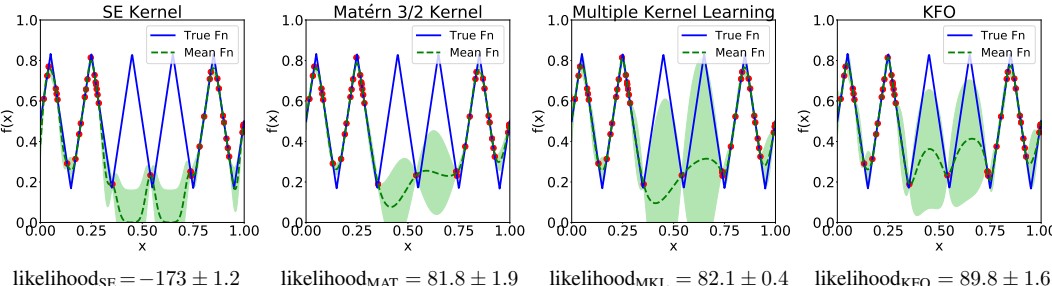

likelihood$_{SE} = -173 \pm 1.2$     likelihood$_{MAT} = 81.8 \pm 1.9$     likelihood$_{MKL} = 82.1 \pm 0.4$     likelihood$_{KFO} = 89.8 \pm 1.6$

Figure 2: Posterior distribution and maximum log-likelihood computed for Triangular wave function using KFO and other baselines. The solid blue line shows the true function. The green shaded area covers two standard deviations above and below the posterior mean shown by the green dashed line.

## 5.2 Real-world Experiments

We compare the performance of our proposed algorithm in SVM classification and GP regression tasks against the state-of-the-art baselines. In our classification and regression experiments, we use the publicly available multi-dimensional real-world datasets from the UCI repository (Dua and Graff, 2017). In SVM classification problems, we use C-SVM in conjunction with KFO to minimise the test classification error ($E_r$). We perform 10-fold cross-validation on the training data set containing 80% of the total instances and tune the cost parameter ($C$) of the SVM in the exponent space of $[-3, 3]$.

We compare our results with Radial Basis Function (RBF) based traditional C-SVM classifier (SVM-RBF) and MKL based SVM classifier (SVM-MKL). We also compare with $\nu$ parameterised Linear SVM ($\nu-$SVM) adhering to the definition of the hyperkernel optimisation problem using the results mentioned in Ong and Smola (2003). The classification error (in %) obtained for the test set consisting of 20% of the total instances using different classifiers over 10 random runs are shown in Table 1. To demonstrate the efficiency of our approach, we also present the best test classification error (last column of Table 1) reported by state-of-the-art classifiers in the literature (Zhang et al., 2017). To the best of our knowledge, Zhang et al. (2017) is the most recent work that surveyed numerous classifiers and reported their performance on UCI datasets. Additionally, we also construct a SVM classifier (KFO-MKL) with its kernel formulated as a weighted combination of KFO tuned kernel and standard kernels (analogous to MKL), we refer to the supplementary material for the results with KFO-MKL.

Table 1: SVM classification results (mean test classification error and its standard deviation) for the real-world datasets using KFO and other baselines, with the test set consisting of 20% of the total instances. Bold indicates the best performance among all the columns. The last column shows the best performance reported among all the other classifiers[†] mentioned in Zhang et al. (2017).

| Dataset | KFO | SVM-RBF | SVM-MKL | $\nu-$SVM | Other Classifiers[†] |
|---|---|---|---|---|---|
| Ionosphere | $\mathbf{3.06 \pm 1.6}$ | $6.1 \pm 1.8$ | $9.31 \pm 0.4$ | $6.7 \pm 1.8$ | $5.56$ |
| Glass | $9.3 \pm 0.19$ | $\mathbf{7.2 \pm 2.7}$ | $20.86 \pm 0.7$ | $8.9 \pm 2.6$ | $9.52$ |
| Sonar | $6.6 \pm 1.2$ | $15.3 \pm 4.1$ | $20.99 \pm 4.4$ | $15.3 \pm 3.7$ | $\mathbf{4.76}$ |
| Heart | $\mathbf{11.47 \pm 0.2}$ | $23.2 \pm 3.7$ | $15$ | $19.3 \pm 2.4$ | $14.14$ |
| Wine | $\mathbf{0}$ | $0.01 \pm 0.01$ | $\mathbf{0}$ | $\mathbf{0}$ | $\mathbf{0}$ |
| Credit | $28.19 \pm 3.57$ | $15.3 \pm 2.0$ | $\mathbf{13.62 \pm 1.8}$ | $13.8 \pm 3.1$ | $24.00$ |
| Biodeg | $12.12 \pm 1.29$ | $26.39 \pm 2.5$ | $12.33 \pm 1.5$ | $14.4 \pm 4.4$ | $\mathbf{11.32}$ |
| Hayes-Roth | $\mathbf{14.18 \pm 0.1}$ | $35.81 \pm 6.2$ | $18.69 \pm 0.9$ | $16.1 \pm 6.9$ | $21.43$ |
| WDBC | $\mathbf{0.68 \pm 0.84}$ | $5.2 \pm 1.4$ | $2.05 \pm 1.6$ | $3.8 \pm 1.2$ | $1.59$ |
| Contraceptive | $\mathbf{30.1 \pm 2.71}$ | $35.49 \pm 1.6$ | $32.86 \pm 2.3$ | $36.95 \pm 3.7$ | $44.59$ |
| Car | $\mathbf{0}$ | $5.30 \pm 0.5$ | $2.54 \pm 0.4$ | $2.22 \pm 0.05$ | $\mathbf{0}$ |
| Phoneme | $22.74 \pm 2.31$ | $22.85 \pm 1.6$ | $20.12 \pm 0.3$ | $17.58 \pm 0.5$ | $\mathbf{10.17}$ |
| Ecoli | $\mathbf{0.96 \pm 0.7}$ | $3.99 \pm 0.7$ | $2.58 \pm 1.3$ | $3.99 \pm 0.7$ | $12.12$ |
| Seeds | $\mathbf{1.2 \pm 0.2}$ | $8.74 \pm 0.4$ | $3.96 \pm 0.5$ | $10.32 \pm 4.8$ | $4.55$ |
| Summary | $9/14$ | $1/14$ | $2/14$ | $1/14$ | $5/14$ |

In GP regression tasks on UCI datasets, we compute the negative log-likelihood (Eq. (1)) on the test set as a measure of performance. We compare our results with the standard parametric kernels such as RBF and Automatic Relevance Determination (ARD) Matérn kernel and the non-parametric kernels such as Functional Kernel Learning based kernels (FKL-Shared and FKL-Separate) mentioned in Benton et al. (2019). In FKL-Separate, the functional kernel learning is achieved by formulating a product of one-dimensional kernels, each of which has its own GP and hyperparameters. In contrast, FKL-Shared uses a GP with unique set of hyperparameters to draw one-dimensional kernels. The results of our GP regression tasks are shown in Table 2, with each cell containing the mean negative log-likelihood and the standard deviation computed over 10 repeated runs with random 80/20 train/test splits. Evidently, our method outperformed the state-of-the-art baselines in both the SVM classification and GP regression experiments, demonstrating the significant improvement in generalisation performance. We refer to the supplementary material for the experimental details and the additional results. The code base used for the experiments mentioned above is available at `https://github.com/mailtoarunkumarav/KernelFunctionalOptimisation`

Table 2: GP Regression results for the real-world datasets using KFO and other baselines. Each cell signifies the mean negative log-likelihood and the standard deviation computed over 10 random runs. Lower the better. Bold indicates the best performance among all the columns.

| Dataset | KFO | RBF | ARD Matérn | FKL-Shared | FKL-Seperate |
|---------|-----|-----|------------|------------|--------------|
| Fertility | $5.15 \pm 2.95$ | $-3.90 \pm 1.76$ | $\mathbf{-4.40 \pm 2.58}$ | $-2.7 \pm 1.25$ | $-1.83 \pm 3.3$ |
| Yacht | $\mathbf{-34.6 \pm 1.6}$ | $1.65 \pm 76.47$ | $-19.49 \pm 14.0$ | $-14.52 \pm 4.7$ | $-15.7 \pm 8.2$ |
| Slump | $\mathbf{-3.01 \pm 1.7}$ | $36.302 \pm 7.93$ | $26.33 \pm 7.48$ | $4.38 \pm 1.33$ | $59.4 \pm 12.87$ |
| Boston | $\mathbf{-24.7 \pm 4.2}$ | $139.61 \pm 11.5$ | $130.8 \pm 10.50$ | $122.6 \pm 3.91$ | $143.7 \pm 5.71$ |
| Auto | $\mathbf{-8.78 \pm 1.2}$ | $96.18 \pm 8.02$ | $94.01 \pm 5.03$ | $101.7 \pm 19.3$ | $98.94 \pm 6.1$ |
| Airfoil | $\mathbf{-204.1 \pm 4.5}$ | $358.93 \pm 8.93$ | $305.88 \pm 7.46$ | $270.0 \pm 28.4$ | $284.8 \pm 48.7$ |

To provide brief insights on the computational time, we have reported the average CPU time (in %) spent optimising (or searching) the kernel and the average CPU time (in %) spent evaluating the kernel by our approach in Table 3. We observe that the percentage of time spent optimising the kernel is no more than $10\%$ of the whole model fitting time. Thus, the proposed method does not add much overhead to the model fitting process. We have also measured the total runtime (in seconds) required for an instance of KFO tuned SVM to complete $S \times T$ iterations, where $S = T = 5$. The total runtime also includes the runtime required for generating 4 random observations in each subspace. The aforesaid runtimes are measured on a server with Intel Xeon processor having 16 GB of RAM.

Furthermore, we ideally expect our proposed method to at least achieve the generalisation performance demonstrated by any standard parametric kernel for the reason that we find the optimum kernel in the whole space of kernels composed of a plethora of kernels including parametric kernels. Although our proposed approach is able to find the global optimal kernel in most cases, we do occasionally observe that our method does not provide the optimal kernel. A possible reason for this could be the insufficient computational budget allocated or the substandard approximations and optimisations. Our empirical results have demonstrated that we can achieve a good generalisation performance even with smaller grids (smaller $N_g$) using Kernel Functional Optimisation (KFO) framework.

Table 3: Runtimes measured for SVM classification using KFO tuned kernel on real-world datasets. The percentage of time spent optimising the kernel and evaluating the kernel is depicted in the fourth and the fifth column, respectively. The total runtime (in seconds) is shown in the last column.

| Dataset | Samples | Features | Optimisation Time(%) | Evaluation Time(%) | Total Runtime |
|---------|---------|----------|----------------------|--------------------|---------------|
| Sonar | 208 | 60 | $9.1 \pm 0.3$ | $90.87 \pm 0.694$ | $567.7 \pm 4.64$ |
| Heart | 303 | 13 | $4.7 \pm 0.35$ | $95.24 \pm 0.567$ | $1070.1 \pm 24.1$ |
| Glass | 214 | 9 | $9.35 \pm 0.55$ | $90.64 \pm 0.441$ | $554.7 \pm 8.19$ |
| Credit | 1000 | 24 | $0.9 \pm 0.13$ | $98.8 \pm 0.360$ | $5191.3 \pm 34.2$ |

## 6 Conclusion

We present a novel formulation for kernel selection via the optimisation of kernel functionals using Bayesian functional optimisation. The kernel functional learnt is a non-parametric kernel capable of capturing the intricate stationary and non-stationary variations. Our algorithm iteratively searches through a sequence of random kernel functional subspaces where the best kernel obtained from all the previous subspace searches biases the next search. The resultant kernel is an indefinite, or *Kreĭn* kernel, thus we use matrix post-processing techniques to ensure the positive definiteness of the resulting Gram matrix. The theoretical analysis shows a fast convergence rate of our algorithm. The experimental results show that our method outperforms the other state-of-the-art baselines.

## Acknowledgments

This research was partially funded by the Australian Government through Australian Research Council (ARC). Prof. Venkatesh is the recipient of an ARC Australian Laureate Fellowship (FL170100006).

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
