# A Appendix

## A.1 Standard Bayesian Optimisation

---

**Algorithm 2** Standard Bayesian Optimisation

---

**Input**: Set of Observations $\widehat{\mathcal{D}}_{1:\hat{t}} = \{\mathbf{x}_{1:\hat{t}}, \mathbf{y}_{1:\hat{t}}\}$

  1. **for** $\hat{t} = 1, 2, \cdots, \widehat{T}$ iterations **do**

  2.        Obtain optimised hyperparameters $\underline{\Theta}^*$ by maximising log marginal likelihood

  3.        Update Gaussian process model with $\underline{\Theta}^*$

  4.        Find $\mathbf{x}_{\hat{t}+1}$ by optimising acquisition function *i.e.,* $\mathbf{x}_{\hat{t}+1} = \operatorname{argmax}_{\mathbf{x} \in \mathcal{X}} u(\mathbf{x})$

  5.        Evaluate the objective function at $\mathbf{x}_{\hat{t}+1}$ *i.e.,* $y_{\hat{t}+1} = f(\mathbf{x}_{\hat{t}+1}) + \varepsilon_{\hat{t}+1}$

  6.        Augment the new observation *i.e.,* $\widehat{\mathcal{D}}_{1:\hat{t}+1} = \widehat{\mathcal{D}}_{1:\hat{t}} \cup (\mathbf{x}_{\hat{t}+1}, y_{\hat{t}+1})$

  7.        Update Gaussian process statistical model

  8. **end for**

---

A standard Bayesian optimisation procedure is mentioned in Algorithm 2. Bayes theorem states that, given the model $\mathcal{M}$ and the data $\widehat{\mathcal{D}}$, the posterior probability of the model given data *i.e.,* $\mathcal{P}(\mathcal{M}|\widehat{\mathcal{D}})$ is proportional to the likelihood of data given model $\mathcal{P}(\widehat{\mathcal{D}}|\mathcal{M})$ multiplied by the prior probability of the model $\mathcal{P}(\mathcal{M})$, mathematically written as

$$\mathcal{P}(\mathcal{M}|\widehat{\mathcal{D}}) \propto \mathcal{P}(\widehat{\mathcal{D}}|\mathcal{M}) \, \mathcal{P}(\mathcal{M})$$

If $y_i = f(\mathbf{x}_i) + \epsilon_i'$ denotes a noisy observation of the unknown objective function for the $i^{\text{th}}$ sample $\mathbf{x}_i$ corrupted with white Gaussian noise ($\epsilon_i'$), then the observation model $\widehat{\mathcal{D}}$ is accumulated as $\widehat{\mathcal{D}}_{1:\hat{t}} = \{\mathbf{x}_{1:\hat{t}}, \mathbf{y}_{1:\hat{t}}\}$. Bayesian optimisation computes the posterior distribution $\mathcal{P}(f|\widehat{\mathcal{D}}_{1:\hat{t}})$ by combining the prior $\mathcal{P}(f)$ with the likelihood $\mathcal{P}(\widehat{\mathcal{D}}_{1:\hat{t}}|f)$ as shown below.

$$\mathcal{P}(f|\widehat{\mathcal{D}}_{1:\hat{t}}) \propto \mathcal{P}(\widehat{\mathcal{D}}_{1:\hat{t}}|f) \, \mathcal{P}(f)$$

The obtained posterior distribution represents our updated belief about the unknown objective function being modelled. There are two main aspects that must be taken into account for the Bayesian optimisation. First, the selection of priors to express our prior belief about the objective function. Gaussian Process (GP) is used for defining prior distributions for the unknown objective function. Second, an acquisition function to determine the next best query point for the function evaluation. Therefore, a standard Bayesian optimisation procedure consists of two main components: (**i**) Gaussian process, and (**ii**) acquisition functions.

### A.1.1 Acquisition Functions

The selection of the next best query point is characterised by an acquisition function. The acquisition function guides the search by balancing the exploration and exploitation trade-off in the input space. We have used Gaussian Process Upper Confidence Bound (GP-UCB) acquisition function (Brochu et al., 2010, Srinivas et al., 2012) in all our experiments at all levels. GP-UCB acquisition function using the upper confidence bound selection criterion is given by

$$u_{GP-UCB}(\mathbf{x}) = \mu(\mathbf{x}) + \sqrt{\beta_{\hat{t}}} \, \sigma(\mathbf{x}) \tag{8}$$

where $\beta_{\hat{t}}$ is a weight that depends on both the number of iterations $\hat{t}$ and the number of observations $\hat{n}$. Srinivas et al. (2012), Brochu et al. (2010) have discussed the characteristics of $\beta_{\hat{t}}$ by setting its value as $\beta_{\hat{t}} = 2\log\left(\frac{\hat{t}^{2+\frac{\hat{n}}{2}}\pi^2}{3\hat{\delta}}\right)$, where $\hat{\delta} \in (0,1)$. In our experiments, we have followed Srinivas et al. (2012), Brochu et al. (2010) to set the value for $\beta_t$ in the inner-loop. The theoretical analysis of the GP-UCB acquisition function (Brochu et al., 2010, Srinivas et al., 2012) demonstrates the better convergence rates when compared to the other popular acquisition functions. Alternatively, we can

use Expected Improvement (EI) acquisition function (Wilson et al., 2018, Močkus, 1975) to guide the search by taking into account the expected improvement over the current optima. If $f(\mathbf{x}^+)$ is the best value observed, then the next best query point is obtained by maximising the EI acquisition function $u_{\text{EI}}(\mathbf{x})$, given by

$$u_{\text{EI}}(\mathbf{x}) = \begin{cases} (\mu(\mathbf{x}) - f(\mathbf{x}^+)) \, \Phi(\mathcal{Z}) + \sigma(\mathbf{x}) \, \phi(\mathcal{Z}) & \text{if } \sigma(\mathbf{x}) > 0 \\ 0 & \text{if } \sigma(\mathbf{x}) = 0 \end{cases}$$

$$\mathcal{Z} = \frac{\mu(\mathbf{x}) - f(\mathbf{x}^+)}{\sigma(\mathbf{x})}$$

where $\Phi(\mathcal{Z})$ and $\phi(\mathcal{Z})$ represents the Cumulative Distribution Function (CDF) and the Probability Density Function (PDF) of the standard normal distribution, respectively.

## A.2 Construction of Matérn Harmonic Hyperkernel

We follow the strategy mentioned in Ong et al. (2003) for the construction of hyperkernels. Let $k$ be a valid positive definite kernel such that $k(\cdot, \cdot) \geq 0$, and $\tilde{\delta} : \mathbb{R} \to \mathbb{R}$ a function that can be represented as a power series with positive coefficients *i.e.*, $\tilde{\delta}(\zeta) = \sum_{\overline{n}=0}^{\infty} \overline{c}_{\overline{n}} \zeta^{\overline{n}}, \overline{c}_i \geq 0$ for all $\overline{n} = 0, \cdots, \infty$. A valid hyperkernel $\kappa$ can be constructed using $\tilde{\delta}$ functions and a kernel $k$ as shown below.

$$\kappa(\tilde{\mathbf{x}}_i, \tilde{\mathbf{x}}_j) := \tilde{\delta}(k(\tilde{\mathbf{x}}_i) k(\tilde{\mathbf{x}}_j)) = \sum_{\overline{n}=0}^{\infty} \overline{c}_{\overline{n}} (k(\tilde{\mathbf{x}}_i) k(\tilde{\mathbf{x}}_j))^{\overline{n}} \tag{9}$$

For example, if $k(\tilde{\mathbf{x}}_i)$ is a Matérn class of covariance functions with $\nu = \frac{3}{2}$ given as

$$k_m(\tilde{\mathbf{x}}) = k_m(\mathbf{x}, \mathbf{x}') = \left(1 + \frac{\sqrt{3}}{l} r\right) \exp\left(-\frac{\sqrt{3}}{l} r\right) \tag{10}$$

where $r = \|\mathbf{x} - \mathbf{x}'\|$, then using the Geometric Maclaurin series expansion of $\frac{1}{1-x}$ and setting the Taylor coefficients $\overline{c}_{\overline{n}}$ to $(1 - \lambda_h) (\lambda_h)^{\overline{n}}$ for some $\lambda_h > 0$ in Eq. (9), we obtain *Matérn Harmonic Hyperkernel* as

$$\kappa(\mathbf{x}, \mathbf{x}', \mathbf{x}'', \mathbf{x}''') = \frac{1 - \lambda_h}{1 - \lambda_h \, k_m(\mathbf{x}, \mathbf{x}') \, k_m(\mathbf{x}'', \mathbf{x}''')}$$

$$\kappa(\mathbf{x}, \mathbf{x}', \mathbf{x}'', \mathbf{x}''') = \frac{1 - \lambda_h}{1 - \left(\lambda_h \, \overline{c}_1 \, \overline{c}_2 \, \exp\left(-\frac{\sqrt{3}}{l}(r_1 + r_2)\right)\right)} \tag{11}$$

where $\lambda_h$ and $l$ correspond to the hyperparameters of the hyperkernel, and $r_1 = \|\mathbf{x} - \mathbf{x}'\|$, $r_2 = \|\mathbf{x}'' - \mathbf{x}'''\|$, $\overline{c}_1 = (1 + \frac{\sqrt{3}}{l} r_1)$, $\overline{c}_2 = (1 + \frac{\sqrt{3}}{l} r_2)$.

As per theory, we can use any universal hyperkernel adhering to the guidelines of hyperkernel framework for defining a Hyper-GP. However, if we have a strong intuition about the properties present in the input space, then we can make use of the explicit hyperkernel construction mentioned in Example 7 of Ong et al. (2003).

## A.3 Kernel Functional Approximation from Hyper-RKHS

The kernel functions $(K)$ used in the Gaussian process uniquely define an associated Reproducing Kernel Hilbert Space (RKHS) $\mathcal{H}_K$ (Aronszajn, 1950). Analogous to the RKHS associated with the kernel function, a hyperkernel $(\kappa)$ defines an associated Hyper-Reproducing Kernel Hilbert Space (Hyper-RKHS) $\mathcal{H}_\kappa$ (Ong et al., 2003). The (hyper) GP distribution $\mathcal{GP}_\kappa(0, \kappa)$ defined by the hyperkernel $\kappa$ is a distribution on the space of kernels. Please see Figure 3 to understand the notion of kernels, hyperkernels, RKHS and Hyper-RKHS.

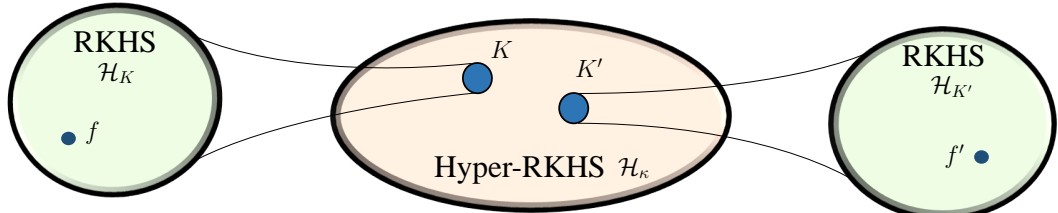

Figure 3: Relation between RKHS defined by kernels and Hyper-RKHS defined by Hyperkernels.

We consider an evenly spaced grid $\mathcal{G}$ with $N_g$ points $\{\tilde{\mathbf{x}}_1, \tilde{\mathbf{x}}_2, \cdots | \tilde{\mathbf{x}}_i \in \tilde{\mathcal{X}} := \mathcal{X} \times \mathcal{X}, \forall i \in \mathbb{N}_{N_g}\}$ for the construction of kernel functionals. Using the grid $\mathcal{G}$, the kernel $K$ can be approximated using the posterior mean of $\mathcal{GP}_\kappa(0, \kappa)$ as

$$K(\tilde{\mathbf{x}}) \approx [\kappa(\tilde{\mathbf{x}}, \tilde{\mathbf{x}}_1)\, \kappa(\tilde{\mathbf{x}}, \tilde{\mathbf{x}}_2)\, \kappa(\tilde{\mathbf{x}}, \tilde{\mathbf{x}}_3)\cdots]\, \boldsymbol{\kappa}^{-1}\, \mathbf{k} = \sum_i \alpha_i\, \kappa(\tilde{\mathbf{x}}, \tilde{\mathbf{x}}_i) \tag{12}$$

where $\boldsymbol{\alpha} = \boldsymbol{\kappa}^{-1}\, \mathbf{k}$. We also note that $\mathbf{k} \sim \mathcal{GP}_\kappa(0, \kappa)$ is essentially a set of noiseless observations of the kernel $K$ on the grid-points $\tilde{\mathbf{x}}_1, \tilde{\mathbf{x}}_2, \cdots$. We expect that the value chosen for the grid size $N_g$ is sufficiently large so that the kernel functionals constructed from the grid is capable of capturing the plentiful properties from the input space. But in practice, it is infeasible to have a very large grid. To avoid the computational burden arising from the larger kernel matrix $\boldsymbol{\kappa} \in \mathbb{R}^{N_g \times N_g}$ resulting from the larger grid size $N_g$, we perform Principal Component Analysis (PCA) (Wold et al., 1987) and choose $N'$ principal components. The Eigen decomposition of the covariance matrix $\boldsymbol{\kappa}$ is given by

$$\boldsymbol{\kappa}\, \widehat{\boldsymbol{E}} = \widehat{\boldsymbol{E}}\widehat{\boldsymbol{\Lambda}}$$
$$\boldsymbol{\kappa} = \widehat{\boldsymbol{E}}\widehat{\boldsymbol{\Lambda}}\widehat{\boldsymbol{E}}^\mathsf{T}$$

where $\widehat{\boldsymbol{E}} \in \mathbb{R}^{N_g \times N_g}$ corresponds to the matrix whose $i^{\text{th}}$ column $\widehat{e}_i$ is the $i^{\text{th}}$ eigenvector of $\boldsymbol{\kappa}$, and $\widehat{\boldsymbol{\Lambda}} \in \mathbb{R}^{N_g \times N_g}$ is the diagonal matrix whose $i^{\text{th}}$ diagonal element $(\widehat{\Lambda}_{ii})$ is the eigenvalue corresponding to the $i^{\text{th}}$ eigenvector. Now, we identify top $N'$ principal components from $\widehat{\boldsymbol{E}}$ and $\widehat{\boldsymbol{\Lambda}}$ to construct $\boldsymbol{E}$ and $\boldsymbol{\Lambda}$ such that

$$\boldsymbol{\kappa} \approx \boldsymbol{E}\boldsymbol{\Lambda}\boldsymbol{E}^\mathsf{T}$$
$$= \boldsymbol{E}\sqrt{\boldsymbol{\Lambda}}(\sqrt{\boldsymbol{\Lambda}})^\mathsf{T}\boldsymbol{E}^\mathsf{T} \tag{13}$$
$$\boldsymbol{\kappa} \approx \boldsymbol{E}\sqrt{\boldsymbol{\Lambda}}(\boldsymbol{E}\sqrt{\boldsymbol{\Lambda}})^\mathsf{T}$$

where $\boldsymbol{E} \in \mathbb{R}^{N_g \times N'}$ corresponds to the matrix whose $i^{\text{th}}$ column $e_i$ is the $i^{\text{th}}$ principal component of $\boldsymbol{\kappa}$, and $\boldsymbol{\Lambda} \in \mathbb{R}^{N' \times N'}$ is the diagonal matrix whose $i^{\text{th}}$ diagonal element $(\Lambda_{ii})$ is the eigenvalue corresponding to the $i^{\text{th}}$ principal component. Therefore, $\boldsymbol{E}\sqrt{\boldsymbol{\Lambda}}$ can be used as a square root of $\boldsymbol{\kappa}$ to construct kernel samples $\mathbf{k}$ as

$$\mathbf{k} = \boldsymbol{E}\sqrt{\boldsymbol{\Lambda}}\boldsymbol{\beta} \qquad \text{where} \quad \boldsymbol{\beta} \sim \mathcal{N}(0, \mathbf{I}_{N'})$$

Further, with the PCA based approximation of $\boldsymbol{\kappa}$ given by Eq. (13), we have $\boldsymbol{\kappa}^{-1} \approx \boldsymbol{E}\boldsymbol{\Lambda}^{-1}\boldsymbol{E}^\mathsf{T}$. Then, the kernel approximation $K(\tilde{\mathbf{x}})$ in Eq. (12) can be simplified as

$$\boldsymbol{\alpha} \approx \boldsymbol{E}\boldsymbol{\Lambda}^{-1}\boldsymbol{E}^\mathsf{T}\boldsymbol{E}\sqrt{\boldsymbol{\Lambda}}\boldsymbol{\beta}$$
$$\approx \boldsymbol{E}\boldsymbol{\Lambda}^{\frac{-1}{2}}\boldsymbol{\beta} \tag{14}$$

As mentioned in the main paper, we fit a GP distribution on the kernel functional observations using the Squared Exponential (SE) kernel ($\overline{k}_{\text{SE}}$) given by

$$\overline{k}_{\text{SE}}(K_1, K_2) = \overline{\sigma}_f^2 \exp\left(\frac{-1}{2\overline{\Upsilon}^2}\|K_1 - K_2\|^2\right) \tag{15}$$

where $\overline{\sigma}_f^2$ and $\overline{\Upsilon}$ correspond to the signal variance and the lengthscale hyperparameters of $\overline{k}_{\text{SE}}$. The similarity between two kernel functionals ($K_1$ and $K_2$) can be approximated using the RKHS norm given as

$$\|K_1 - K_2\|_{\mathcal{H}_{\kappa' \neq \kappa}} \approx \|\mu_1 - \mu_2\|_{\mathcal{H}_\kappa} \tag{16}$$

where $\mu_1$ and $\mu_2$ are the posterior mean approximations of $K_1$ and $K_2$, respectively. The RKHS norm of the posterior mean difference (Eq. (16)) can be computed as shown below.

$$\|\mu_1 - \mu_2\|_{\mathcal{H}_\kappa} = \sqrt{(\boldsymbol{\alpha}_1 - \boldsymbol{\alpha}_2)^\mathsf{T} \, \boldsymbol{\kappa} \, (\boldsymbol{\alpha}_1 - \boldsymbol{\alpha}_2)} \\ \sqrt{\boldsymbol{\alpha}_1^\mathsf{T} \boldsymbol{\kappa} \boldsymbol{\alpha}_1 + \boldsymbol{\alpha}_2^\mathsf{T} \boldsymbol{\kappa} \boldsymbol{\alpha}_2 - 2\boldsymbol{\alpha}_1^\mathsf{T} \boldsymbol{\kappa} \boldsymbol{\alpha}_2} \tag{17}$$

On further simplification with the Eigen decomposition provided in Eq. (14), we get

$$\begin{aligned} \boldsymbol{\alpha}_1^\mathsf{T} \boldsymbol{\kappa} \boldsymbol{\alpha}_1 &= (\boldsymbol{E} \boldsymbol{\Lambda}^{\frac{-1}{2}} \boldsymbol{\beta}_1)^\mathsf{T} \boldsymbol{E} \boldsymbol{\Lambda} \boldsymbol{E}^\mathsf{T} \boldsymbol{E} \boldsymbol{\Lambda}^{\frac{-1}{2}} \boldsymbol{\beta}_1 \\ &= \boldsymbol{\beta}_1^\mathsf{T} \boldsymbol{\Lambda}^{\frac{-1}{2}} \boldsymbol{\Lambda} \boldsymbol{\Lambda}^{\frac{-1}{2}} \boldsymbol{\beta}_1 \\ &= \boldsymbol{\beta}_1^\mathsf{T} \boldsymbol{\beta}_1 \end{aligned} \tag{18}$$

Similarly,

$$\begin{aligned} \boldsymbol{\alpha}_2^\mathsf{T} \boldsymbol{\kappa} \boldsymbol{\alpha}_2 &= \boldsymbol{\beta}_2^\mathsf{T} \boldsymbol{\beta}_2, \text{and} \\ 2\boldsymbol{\alpha}_1^\mathsf{T} \boldsymbol{\kappa} \boldsymbol{\alpha}_2 &= 2 \cdot \boldsymbol{\beta}_1^\mathsf{T} (\boldsymbol{\Lambda}^{\frac{-1}{2}})^\mathsf{T} \boldsymbol{\Lambda} \boldsymbol{\Lambda}^{\frac{-1}{2}} \boldsymbol{\beta}_2 \\ &= 2 \cdot \boldsymbol{\beta}_1^\mathsf{T} \boldsymbol{\beta}_2 \end{aligned} \tag{19}$$

Using Eq. (18) and Eq. (19) in the RKHS norm of posterior mean difference (Eq. (17)), we obtain

$$\|K_1 - K_2\|_{\mathcal{H}_{\kappa' \neq \kappa}} = \sqrt{\boldsymbol{\beta}_1^\mathsf{T} \boldsymbol{\beta}_1 + \boldsymbol{\beta}_2^\mathsf{T} \boldsymbol{\beta}_2 - 2 \cdot \boldsymbol{\beta}_1^\mathsf{T} \boldsymbol{\beta}_2} \tag{20}$$

Alternatively, the similarity between kernel functionals mentioned in Eq. (15) can be approximated using $L_2-$Norm given as

$$\|K_1 - K_2\|_{L_2} \approx \|\mathbf{k}_1 - \mathbf{k}_2\| = \sqrt{\mathbf{k}_1^\mathsf{T} \mathbf{k}_1 + \mathbf{k}_2^\mathsf{T} \mathbf{k}_2 - 2 \cdot \mathbf{k}_1^\mathsf{T} \mathbf{k}_2}$$

where

$$\begin{aligned} \mathbf{k}_1^\mathsf{T} \mathbf{k}_1 &= (\boldsymbol{E} \sqrt{\boldsymbol{\Lambda}} \boldsymbol{\beta}_1)^\mathsf{T} \boldsymbol{E} \sqrt{\boldsymbol{\Lambda}} \boldsymbol{\beta}_1 \\ &= \boldsymbol{\beta}_1^\mathsf{T} \sqrt{\boldsymbol{\Lambda}}^\mathsf{T} \boldsymbol{E}^\mathsf{T} \boldsymbol{E} \sqrt{\boldsymbol{\Lambda}} \boldsymbol{\beta}_1 \\ &= \boldsymbol{\beta}_1^\mathsf{T} \boldsymbol{\Lambda} \boldsymbol{\beta}_1 \\ \mathbf{k}_1^\mathsf{T} \mathbf{k}_2 &= (\boldsymbol{E} \sqrt{\boldsymbol{\Lambda}} \boldsymbol{\beta}_1)^\mathsf{T} \boldsymbol{E} \sqrt{\boldsymbol{\Lambda}} \boldsymbol{\beta}_2 \\ &= \boldsymbol{\beta}_1^\mathsf{T} \sqrt{\boldsymbol{\Lambda}}^\mathsf{T} \boldsymbol{E}^\mathsf{T} \boldsymbol{E} \sqrt{\boldsymbol{\Lambda}} \boldsymbol{\beta}_2 \\ &= \boldsymbol{\beta}_1^\mathsf{T} \boldsymbol{\Lambda} \boldsymbol{\beta}_2 \end{aligned}$$

and consequently,

$$\|K_1 - K_2\|_{L_2} \approx \sqrt{\boldsymbol{\beta}_1^\mathsf{T} \boldsymbol{\Lambda} \boldsymbol{\beta}_1 + \boldsymbol{\beta}_2^\mathsf{T} \boldsymbol{\Lambda} \boldsymbol{\beta}_2 - \mathbf{2} \cdot \boldsymbol{\beta}_1^\mathsf{T} \boldsymbol{\Lambda} \boldsymbol{\beta}_2} \tag{21}$$

In all our experiments, we compute the similarity between kernel functionals using the RKHS norm mentioned in Eq. (16).

### A.4   Theoretical Analysis

### A.4.1   Proof of Proposition 1

Our proof of Proposition 1 is pivoted on extending the concept of information gain $\gamma_t$ (Cover, 1999) in a standard Euclidean space to a functional space of kernels. Our model in the inner-loop is built directly on the projections of the variables ($\boldsymbol{\lambda}^*$) optimised in the functional space of kernels *i.e.,*

$\mathbf{k} = \mathbf{k}^{\#} + \sum_i \lambda^{(i)} \mathbf{k}^{(i)}$ (we use $K$ and $\mathbf{k}$ interchangeably here as both represent the kernel as per Eq. (12)). For each instantiation $s$ of the outer-loop, the corresponding subspace is constructed as $\mathbb{S}_s = \mathbf{k}^{\#} + \text{span}(\mathbf{k}^{(1)} + \cdots + \mathbf{k}^{(d)})$, where $\mathbf{k}^{\#}$ is the optimal kernel found in the previous $s-1$ subspace searches. Therefore, $\forall K_1, K_2 \in \mathbb{S}_s$, the RKHS norm $\|K_1 - K_2\|_{\mathcal{H}_{\kappa' \neq \kappa}}$ required for computing the similarity $\overline{k}(K_1, K_2)$ between kernel functionals can be written as shown below.

$$
\begin{aligned}
\|K_1 - K_2\|_{\mathcal{H}_{\kappa' \neq \kappa}} &\approx \big\| \sum_i (\lambda_1^{(i)} - \lambda_2^{(i)}) \mathbf{k}^{(i)} \big\|_{\mathcal{H}_{\kappa'}} \\
&= (\boldsymbol{\lambda}_1 - \boldsymbol{\lambda}_2)^{\intercal} M (\boldsymbol{\lambda}_1 - \boldsymbol{\lambda}_2)
\end{aligned}
\tag{22}
$$

where $M \succeq 0$, $M_{pq} = \langle \mathbf{k}^{(p)}, \mathbf{k}^{(q)} \rangle$, having the form of Mahalanobis distance. The new formulation is identical to the standard Euclidean distance measured on transformed data with appropriate scaling and rotation. Interestingly, the maximum information gain $\gamma_t$ of a kernel at iteration $t$ of the inner-loop depends only on the number of observations, irrespective of the inherent transformations such as linear transformation (scaling), rotation, *etc.* on the datapoints. Thus, the kernel construction using a translation-invariant covariance allows the kernel $\overline{k} : \mathbb{S}_s \times \mathbb{S}_s \to \mathbb{R}$ in functional space to share the same maximum information gain $\gamma_t$ as that of the standard kernel $k : \mathcal{X} \times \mathcal{X} \to \mathbb{R}$ in the Euclidean space (see Seeger et al. (2008) and Srinivas et al. (2012) for the maximum information gain of standard kernels).

$\blacksquare$

### A.4.2 Proof of Theorem 1

The proof of Theorem 1 builds on deriving the maximum information gain ($\gamma_t$) rate for the kernels in biased subspaces. In the inner-loop, the model for $f$ is constructed using the observations accumulated from the current subspace $s$, as well as the previous $s-1$ subspaces. Therefore, the posterior of $f$ in subspace $s$ before proceeding with the inner-loop is $f(K)|\mathcal{D}_{s-1} \sim \mathcal{N}(\mu_{\mathcal{D}_{s-1}}(K), \sigma^2_{\mathcal{D}_{s-1}}(K))$, where $\mathcal{D}_{s-1}$ denotes the set of all observations obtained from the previous $s-1$ subspaces. Further, the posterior of $f$ at iteration $t$ of the inner-loop is given as $f(K)|\mathcal{D}_{s-1} \cup \mathcal{D}'_s \sim \mathcal{N}(\mu_{\mathcal{D}_{s-1} \cup \mathcal{D}'_s}(K), \sigma^2_{\mathcal{D}_{s-1} \cup \mathcal{D}'_s}(K))$. It can be shown that the updated posterior $f(K)|\mathcal{D}_{s-1} \cup \mathcal{D}'_s$ is equivalent to the posterior of the biased GP having the prior covariance $\widehat{k}_{\mathcal{D}_{s-1}}$ *i.e.*, $f \sim \mathcal{GP}(\mu_{\mathcal{D}_{s-1}}, \widehat{k}_{\mathcal{D}_{s-1}})$ induced by observations $\mathcal{D}_{s-1}$ from the previous subspace searches. The posterior distribution for the function $f \sim \mathcal{GP}(0, \widehat{k})$ given the observations from the current and the previous subspace searches is given by

$$
\mu_{\mathcal{D}_{s-1} \cup \mathcal{D}'_s}(K) = \begin{bmatrix} \widehat{k}(\mathcal{D}_{s-1}, K) \\ \widehat{k}(\mathcal{D}'_s, K) \end{bmatrix}^{\intercal} \begin{bmatrix} \widehat{k}(\mathcal{D}_{s-1}, \mathcal{D}_{s-1}) + \sigma_n^2 \mathbf{I} & \widehat{k}(\mathcal{D}_{s-1}, \mathcal{D}'_s) \\ \widehat{k}(\mathcal{D}'_s, \mathcal{D}_{s-1}) & \widehat{k}(\mathcal{D}'_s, \mathcal{D}'_s) + \sigma_n^2 \mathbf{I} \end{bmatrix}^{-1} \begin{bmatrix} \mathbf{y}_{\mathcal{D}_{s-1}} \\ \mathbf{y}_{\mathcal{D}'_s} \end{bmatrix}
\tag{23}
$$

$$
\sigma^2_{\mathcal{D}_{s-1} \cup \mathcal{D}'_s}(K) = \widehat{k}(K, K) - \begin{bmatrix} \widehat{k}(\mathcal{D}_{s-1}, K) \\ \widehat{k}(\mathcal{D}'_s, K) \end{bmatrix}^{\intercal} \begin{bmatrix} \widehat{k}(\mathcal{D}_{s-1}, \mathcal{D}_{s-1}) + \sigma_n^2 \mathbf{I} & \widehat{k}(\mathcal{D}_{s-1}, \mathcal{D}'_s) \\ \widehat{k}(\mathcal{D}'_s, \mathcal{D}_{s-1}) & \widehat{k}(\mathcal{D}'_s, \mathcal{D}'_s) + \sigma_n^2 \mathbf{I} \end{bmatrix}^{-1} \begin{bmatrix} \widehat{k}(\mathcal{D}_{s-1}, K) \\ \widehat{k}(\mathcal{D}'_s, K) \end{bmatrix}
\tag{24}
$$

Similarly, the posterior distribution of $f$ given only the observations from previous subspaces search is represented as shown below.

$$
\mu_{\mathcal{D}_{s-1}}(K) = \widehat{k}(K, \mathcal{D}_{s-1}) [\widehat{k}(\mathcal{D}_{s-1}, \mathcal{D}_{s-1}) + \sigma_n^2 \mathbf{I}]^{-1} \mathbf{y}_{\mathcal{D}_{s-1}}
\tag{25}
$$

$$
\sigma^2_{\mathcal{D}_{s-1}}(K) = \widehat{k}(K, K) - \widehat{k}(K, \mathcal{D}_{s-1}) [\widehat{k}(\mathcal{D}_{s-1}, \mathcal{D}_{s-1}) + \sigma_n^2 \mathbf{I}]^{-1} \widehat{k}(\mathcal{D}_{s-1}, K)
\tag{26}
$$

$$
\widehat{k}_{\mathcal{D}_{s-1}}(K, K') = \widehat{k}(K, K') - \widehat{k}(K, \mathcal{D}_{s-1}) [\widehat{k}(\mathcal{D}_{s-1}, \mathcal{D}_{s-1}) + \sigma_n^2 \mathbf{I}]^{-1} \widehat{k}(\mathcal{D}_{s-1}, K')
\tag{27}
$$

It is clearly seen that the posterior distribution mentioned in Eq. (23) and Eq. (24) can be reformulated in terms of Eq. (25) and Eq. (26) using matrix inversion lemma. Therefore, the final posterior mean

and variance given the observations from the current and previous subspace searches are computed as

$$\mu_{\mathcal{D}_{s-1}\cup\mathcal{D}'_s}(K) = \mu_{\mathcal{D}_{s-1}}(\mathcal{D}'_s)+\widehat{k}_{\mathcal{D}_{s-1}}(K,\mathcal{D}'_s)[\widehat{k}_{\mathcal{D}_{s-1}}(\mathcal{D}'_s,\mathcal{D}'_s)+\sigma_n^2\mathbf{I}]^{-1}(\mathbf{y}_{\mathcal{D}_{s-1}}-\mu_{\mathcal{D}_{s-1}}(\mathcal{D}'_s)) \quad (28)$$

$$\sigma^2_{\mathcal{D}_{s-1}\cup\mathcal{D}'_s}(K) = \widehat{k}_{\mathcal{D}_{s-1}}(K,K) - \widehat{k}_{\mathcal{D}_{s-1}}(K,\mathcal{D}'_s)[\widehat{k}_{\mathcal{D}_{s-1}}(\mathcal{D}'_s,\mathcal{D}'_s) + \sigma_n^2\mathbf{I}]^{-1}\widehat{k}_{\mathcal{D}_{s-1}}(\mathcal{D}'_s,K) \quad (29)$$

Therefore, the updated posterior $f(K)|\mathcal{D}_{s-1} \cup \mathcal{D}'_s$ is equivalent to the posterior of the biased GP with prior covariance $\widehat{k}_{\mathcal{D}_{s-1}}$ (Eq. (27)) *i.e.,* $f \sim \mathcal{GP}(\mu_{\mathcal{D}_{s-1}}, \widehat{k}_{\mathcal{D}_{s-1}})$ induced by observations $\mathcal{D}_{s-1}$ from the previous subspace searches.

Hence, the inner-loop will have sub-linear regret as per GP-UCB based BO (see Theorem 2 of Srinivas et al. (2012)), but with prior covariance $\widehat{k}_{\mathcal{D}_{s-1}}$ (as per Eq. (27)). The maximum information gain for this covariance at any instance of the inner-loop is denoted by $\gamma_{\mathcal{D}_{s-1},t}$. Therefore, the regret for the inner-loop with SE kernel grows as $\mathcal{O}^*(\sqrt{dt\gamma_{\mathcal{D}_{s-1},t}})$, where $\mathcal{O}^*$ notation corresponds to the variant of $\mathcal{O}$ , but with log factors suppressed. For any subspace $s$, till the iteration $t$, the maximum information gain is $\gamma_{\mathcal{D}_{s-1},t} = \gamma_{sT+t}$, and hence the regret bounds for the SE kernel ($\bar{k}_{\mathrm{SE}}$) can be obtained as $\gamma_{\mathcal{D}_{s-1},t} \in \mathcal{O}((\log t)^{d+1})$. Consequently, the overall regret bound for the inner-loop is given by $\mathcal{O}(\sqrt{dt(\log t)^{d+1}})$. For the other standard kernels, we just need to use their corresponding maximum information gain rate. The maximum information gain rate for the other standard kernels is discussed in Seeger et al. (2008), Srinivas et al. (2012). ∎

### A.4.3 Proof of Theorem 2

Our proof of Theorem 2 follows the analysis mentioned in Kirschner et al. (2019), Shilton et al. (2020), but with a hyperkernel perspective and slight modifications in terms of the number of effective dimensions and the varying upper bounds. We define some notations for the convenience of the readers. Let $\lfloor t \rfloor$ in the subscript of any variable denote "before the iteration $t$", whereas a plain-language $t$ in the subscript denote "for the iteration $t$". For example, $\mathcal{D}_{\lfloor t \rfloor}$ denotes the set of all observations before proceeding with iteration $t$, whereas $\mathcal{D}_t$ is the set of observations in iteration $t$. First, we provide some preliminary results to support our theoretical proof. Let $\mathfrak{C}_{\overline{\kappa},d}(\mathfrak{o})$ be a distribution of random projections of $\mathcal{H}_{\overline{\kappa}}$ spanned by $\mathfrak{o} + \mathrm{span}(\mathbf{k}^{(1)}, \mathbf{k}^{(2)}, \cdots, \mathbf{k}^{(d)})$ with fixed origin $\mathfrak{o}$ and $\mathbf{k}^{(1)}, \mathbf{k}^{(2)}, \cdots, \mathbf{k}^{(d)} \sim \mathcal{GP}_\kappa(0, \kappa)$. For each instantiation of the outer-loop ($s$) in Algorithm 1 (mentioned in the main paper), the subspace considered for the optimisation is denoted as $\mathbb{S}_s \sim \mathfrak{C}_{\overline{\kappa},d}(\mathfrak{o}_\mathbf{s})$. Let $\mathbb{S}_{\lfloor s \rfloor} = \bigcup_{i\in\mathbb{N}_s} \mathbb{S}_i$, where $\mathbb{N}_s = \{1, \cdots, s\}$. Let $\mathbf{x}_s^* = \underset{\mathbf{x}\in\mathbb{S}_s}{\mathrm{argmax}}\, f(\mathbf{x})$ and $\mathbf{x}_{\lfloor s \rfloor}^* = \underset{\mathbf{x}\in\mathbb{S}_{\lfloor s \rfloor}}{\mathrm{argmax}}\, f(\mathbf{x})$. First, we prove the following lemma to supplement our results.

**Lemma 1:** *Let $\mathbb{S} \subseteq \mathbb{V} = \{\mathbf{p} \in \mathbb{R}^{d'} | \|\mathbf{p}\|_{L_2} \leq L\}$, where $\mathbb{S}$ is defined by $\mathrm{span}(\mathbf{s}^1, \mathbf{s}^2, \cdots, \mathbf{s}^d) + \mathbf{b} \bigcap \mathbb{V}$, $\mathbf{s}^i \perp \mathbf{s}^j \,\forall i \neq j \in \mathbb{N}_d$, $\mathbf{s}^i \sim \mathcal{S}_i \,\forall i \in \mathbb{N}_d$, $\mathbf{b} \sim \mathcal{B}$ for some distributions $\mathcal{S}_i$ and $\mathcal{B} \in \mathbb{V}$. Then the probability that $\mathbb{S}$ intersects the $d'$ ball of radius $r = \rho L$, where $\rho \in (0, 1]$, at the origin is at least $\Omega(\rho^{d'-d})$ if $d < d'$, 1 otherwise.*

Proof: We are reproducing the results here for the sake of completeness. Let $\Xi_{d,d'}$ be the probability of intersection. We know that $\|\mathbf{s}^i\|_{L_2}^2 \neq 0 \,\forall i \in \mathbb{N}_d$. Therefore, we assume $\dim(\mathbb{S}) = d$. If $d = d'$ then $\mathbb{S} = \mathbb{V}$ and $\Xi_{d,d'} = 1$. Furthermore, if $d = 0$, then the probability that it intersects is equal to the probability that a random point chosen from a distribution falls into a $d'$ ball (of radius $\rho L$), which is defined as the ratio of the measure of the $d'$ ball to the measure of $\mathbb{V}$ *i.e.,* $\Xi_{0,d'}(\rho) = \Omega(\rho^{d'})$. On the other hand, if $0 < d < d'$, as both $d'$ ball with radius $r = \rho L$ and $\mathbb{V}$ are invariant to the rotations about origin, we thus assume that $\mathbf{s}^i = [\overline{\delta}_{1,i} \, \overline{\delta}_{2,i} \, \cdots \, \overline{\delta}_{d,i} \, \mathbf{0}]\overline{s}^i$, where $i \in \mathbb{N}_d$ and $\overline{\delta}_{i,j}$ is the Kronecker-delta. Hence, for all $\overline{s}^i \neq 0$, $\mathbf{v} = [\hat{\mathbf{v}} \, \check{\mathbf{v}}], \hat{\mathbf{v}} \in \mathbb{R}^d, \check{\mathbf{v}} \in \mathbb{R}^{d'-d}$ and $\check{\mathbf{v}} \sim \tilde{\mathcal{B}}$, we have

$$\Xi_{d,d'}(\rho) = \mathcal{P}\left(\min_{\tilde{\mathbf{v}}\in\mathbb{R}^d}\left\|\begin{bmatrix}\hat{\mathbf{v}} + \tilde{\mathbf{v}}\odot\overline{s} \\ \check{\mathbf{v}}\end{bmatrix}\right\|_{L_2} \leq \rho L\right)$$
$$= \mathcal{P}(\|\check{\mathbf{v}}\|_{L_2} \leq \rho L) \quad (30)$$

where $\odot$ is the Hadamard product. The minimum of the probability mentioned in Eq. (30) is obtained when $\tilde{v}_i = -\frac{\hat{v}_i}{\bar{s}^i} \ \forall i \in \mathbb{N}_d$, which is exactly the case corresponding to $d = 0$ with decreased $d'$. Therefore, $\Xi_{d,d'}(\rho) = \Xi_{0,d'-d}(\rho) = \Omega(\rho^{d'-d})$ if $0 < d < d'$. $\blacksquare$

**Note:** The orthogonality condition in Lemma 1 is not a necessary condition. The key point here is that $\mathbb{S}$ is a random $d$-dimensional subspace intersecting a ball of radius $L$ in $d'$-dimensional space ($d \leq d'$). The vectors $\mathbf{s}^1, \mathbf{s}^2, \ldots \mathbf{s}^d$ randomly drawn from a smooth distribution are almost surely linearly independent and thus define a (non-orthogonal) basis for the subspace $\mathbb{S}$ of dimension $d$. Starting from such a basis, we can then obtain an orthogonal basis for $\mathbb{S}$ using the Gram-Schmidt procedure, and thus the proof of Lemma 1 holds.

**Lemma 2:** *For any instantiation (s) of the outer-loop, $\mathcal{P}(f(K^*) - f(K^*_{\downarrow s \downarrow}) \leq \tau) \geq 1 - \exp(-s\varrho(\tau))$, where $\varrho(\tau)$ is a lower bound: $\varrho(\tau) \leq \mathcal{P}(\exists K \in \mathbb{S}, f(K^*) - f(K) \leq \tau | \mathbb{S} \in \mathfrak{C}_{\overline{\kappa},d}(\mathfrak{o}))$. Further, if the first-order minimum condition is met at $K^*$ then $\varrho(\tau) = \Omega(\tau^{\frac{d'-d}{2}})$ if $d < d'$, 1 otherwise.*

Proof: We follow the proof of Lemma 2 in Shilton et al. (2020) to theoretically extend our Theorem 2 to the domain of kernel functionals defined by hyperkernel $\kappa$. Using the inequality $1 - x \leq e^{-x}$,

$$\begin{aligned}
\mathcal{P}(f(K^*) - f(K^*_{\downarrow s \downarrow}) \leq \tau) &= 1 - \mathcal{P}(f(K^*) - f(K^*_{\downarrow s \downarrow}) \geq \tau) \\
&\geq 1 - \prod_{i \in \mathbb{N}_s} \mathcal{P}(f(K^*) - f(K^*_i) \geq \tau) \\
&\geq 1 - (1 - \varrho(\tau))^s \\
&\geq 1 - \exp(-s\varrho(\tau))
\end{aligned} \tag{31}$$

By definition we know that, there exists $\mathcal{K}^1, \mathcal{K}^2, \cdots, \mathcal{K}^{d'} \in \mathcal{H}_\kappa$ such that $\|f(K_\top + K_\perp) - f(K_\top)\|_{\mathcal{H}_\kappa} = 0 \quad \forall K_\top \in \mathbb{K}, \forall K_\perp \in \mathbb{K}^\perp$, where $\mathbb{K} = \mathrm{span}(\mathcal{K}^1, \mathcal{K}^2, \cdots, \mathcal{K}^{d'})$ and $\mathbb{K}^\perp = \{\tilde{K} \in \mathcal{H}_\kappa \mid \langle K, \tilde{K} \rangle_{\mathcal{H}_\kappa} = 0, \forall K \in \mathbb{K}\}$. Further, $\forall K \in \mathcal{H}_\kappa$, $K = K_\top + K_\perp$, we define the set of best solutions at a distance $\tau$ from the optima as, $\mathbb{V}_\tau = \{K \in \mathcal{H}_\kappa | f(K^*) - f(K) \leq \tau\}$. Also, $f(K) = f(K_\top)$, therefore $\mathbb{V}_\tau = \mathbb{V}_{\tau\top} \bigoplus \mathbb{K}^\perp$, where $\mathbb{V}_{\tau\top} = \{K_\top \in \mathbb{K} | f(K^*) - f(K_\top) \leq \tau\}$ is said to have $d'$ dimensions. Thus, bounding the probability that a random $d$-dimensional subspace $\mathbb{S} \sim \mathfrak{C}_{\overline{\kappa},d}(\mathfrak{o})$ projected onto $\mathbb{K}$ intersects $\mathbb{V}_{\tau\top}$ further places a lower bound on $\varrho(\tau)$. To achieve this, we define $\tilde{\mathbb{V}}_{\tau,\eta\top} = \{K_\top \in \mathbb{K} | \frac{\eta}{2L_{\max}^2} \|K^* - K_\top\|_{\mathcal{H}_\kappa}^2 \leq \tau\}$, where $\eta > 0$. Since $f$ is twice Frechet-differentiable, we find that, for small $\frac{\eta}{2L_{\max}^2} \|\overline{\mathbf{w}}\|_{\mathcal{H}_\kappa}^2$, we have $f(K^* + \overline{\mathbf{w}}) \geq f(K^*) - \frac{\eta}{2L_{\max}^2} \|\overline{\mathbf{w}}\|_{\mathcal{H}_\kappa}^2$. If $\overline{\mathbf{w}} = K_\top - K^*$, we see that $f(K^*) - f(K_\top) \leq \frac{\eta}{2L_{\max}^2} \|K_\top - K^*\|_{\mathcal{H}_\kappa}^2$, proving $\tilde{\mathbb{V}}_{\tau,\eta\top} \subseteq \mathbb{V}_{\tau\top}$. Therefore, a lower bound on $\varrho(\tau)$ can be ensured by bounding the probability that a subspace ($\mathbb{S}_\top$) with $d$-dimensions lying in a $d'$ dimensional subspace intersects a $d'$ ball with radius $r = \sqrt{\frac{2\tau}{\eta}} L_{\max}$ at the origin. By the proof provided in Lemma 1, the aforesaid probability is $\Omega(\tau^{\frac{d'-d}{2}})$ if $d < d'$ and 1 otherwise, thereby proving Lemma 2. $\blacksquare$

The regret bounds for the inner-loop is established using the results from Proposition 1 and Theorem 1. Further, the regret bounds ($\epsilon_{d,\delta}$) for the inner-loop depends on the way in which the inner-loop is terminated *i.e.,* after a fixed number of iterations or until the accepted solution accuracy is attained (see discussion in Section 4.3 of our main paper for the overall convergence of the algorithm).

d-dimensional hyperplane

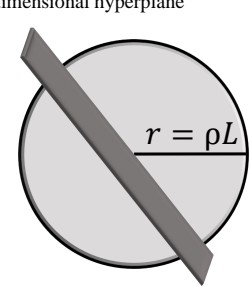

$r = \rho L$

Figure 4: $d-$ dimensional hyperplane intersecting $d'$ ball of radius $r = \rho L$.

With the aforestated results, our proof of Theorem 2 follows the proof of Proposition 1 provided in Kirschner et al. (2019), except that our upper bound varies in the order $\epsilon_{d,\delta}$, instead of having a fixed upper bound $\epsilon$ and we have $d' - d$ instead of $d' - 1$ in the overall regret bound. We note that the proof of Proposition 1 in Kirschner et al. (2019) discusses a special case of our Theorem 2, where **(i)** $d = 1$ (line-search), **(ii)** the search space is finite-dimensional (ours is infinite-dimensional (functional) with finite effective dimensions), **(iii)** it is assumed that the inner-loop runs until $\text{err}(\mathbf{x})$ (Eq. (4) in Kirschner et al. (2019)) drops below a fixed threshold. If we start with the proof of Proposition 1 in Kirschner et al. (2019), replace their Lemma 2 with ours to address point **(i)**, note that the effective dimension is the dimension relevant to the proof to address point **(ii)**, and make $\epsilon$ (in our case) explicitly dependent on $d$ and $\delta$ *i.e.,* $\epsilon_{d,\delta}$ to take care of point **(iii)** (discussed in Sec. 4.3 of our main paper), we obtain a proof of our Theorem 2.

### A.5 Experimental Setup and Additional Results

#### A.5.1 Parameter Selection in KFO

The parameter $S$ in Algorithm 1 corresponds to the number of subspace searches and the parameter $T$ corresponds to the number of iterations in each of those subspaces. In this work, we have considered a fixed computational budget ($S \times T$) for the optimisation and set $S = 5$ and $T = 20$. If we are not concerned with the computational budget, we can set up a stopping criterion (following Kirschner et al. (2019)) based on the quality of the solution achieved *i.e.,* the terminating condition is set with a measure such as simple regret or cumulative regret to keep track of the optimisation performance. We have discussed the convergence guarantees of our approach for both the criteria (fixed and variable $T$) in Theoretical Analysis section.

The number of points $N_g$ in the grid $\mathcal{G}$ is an important parameter. A very fine resolution grid ensures that we can capture small-scale patterns in the kernel. However, a large grid size comes with large computational costs. As a trade-off we set the grid size *i.e.,* $N_g \geq 10 \times n$, where $n$ is the dimension of the given problem, and we find that this works in practice (specifically, (a) it outperforms the baselines in majority of the datasets, and (b) we observe that increasing $N_g$ further does not give significant improvements in performance). Our empirical results justify that the current assumptions made on the grid is fair enough to arrive at optimal solutions. To avoid the slower training time arising from the larger grids, we are planning to explore in the direction of Random Fourier Features for the efficient computation as part of our future line of work.

As we deal with the kernel selection problem, the dataset-specific characteristics will have a minimal impact on the optimum kernel that can be obtained using Kernel Functional Optimisation (KFO). This is true for most of the Bayesian optimisation literature, where the convergence is not tightly coupled with the dataset-specific attributes. In contrast, the convergence heavily depends on the complexity of the observed search space, in our case the search space is a space of kernels defined by placing a GP distribution on kernels using hyperkernel $\kappa$. Further, our proposed approach KFO treats any dataset equally and the dataset need not originate from the UCI repository always. Thus, the strategy to select the computational budget and other parameters remains the same for any class of the dataset.

**Hyperparameter Tuning**    As discussed in the main paper, we fit a GP distribution $\mathcal{GP}(0, \overline{k}_{\text{SE}})$ on the observed kernel functionals in the inner-loop. Although any kernel is equally valid for fitting a GP distribution on kernel functional observations, we have considered the commonly used Squared Exponential (SE) Kernel for our experiments. Further, we provide the convergence guarantees of our algorithm considering the SE kernel in the inner-loop of our algorithm. The hyperparameter set $\overline{\theta} = \{\overline{\sigma}_f^2, \overline{\Upsilon}\}$ of the SE kernel are tuned by maximising the GP log marginal likelihood given as

$$\mathcal{L} = \mathcal{P}(\mathbf{y}|X, \Theta) = \int \mathcal{P}(\mathbf{y}|\mathbf{f})\, \mathcal{P}(|X, \Theta)\, df \tag{32}$$

The closed-form formulation for GP log marginal likelihood is given as

$$\log \mathcal{L} = -\frac{1}{2}(\mathbf{y}^{\mathsf{T}}[\mathbf{C} + \sigma_{noise}^2 \mathbf{I}]^{-1}\mathbf{y}) - \frac{1}{2}\log|\mathbf{C} + \sigma_{noise}^2 \mathbf{I}| - \frac{\check{n}}{2}\log(2\pi) \tag{33}$$

where $C$ corresponds to the covariance matrix constructed for $\check{n}$ observations. The hyperparameters $\overline{\sigma}_f^2$ and $\overline{\Upsilon}$ of the SE kernel in the inner-loop are tuned in the interval $(0, 5]$ and $(0, 1]$, respectively.

Additionally, to achieve better kernel approximations at Hyper-GP level, the hyperparameters $\Theta = \{\lambda_h, l\}$ of Matérn harmonic hyperkernel $\kappa$ mentioned in Eq. (11) are tuned using a separate standard Bayesian optimisation procedure (Algorithm 2). The hyperparameters $\lambda_h$ and $l$ are tuned in the interval $(0, 1]$ and $(0, 1]$, respectively. In this Bayesian optimisation procedure, the observation model is constructed as $\mathfrak{D} = \{(\Theta, y' = \Gamma(\Theta))\}$, where $\Gamma$ maps the model performance $y'$ with the corresponding hyperparameter set $\Theta$. Then, we fit a GP surrogate model $\mathcal{GP}_H$ on the observations $\mathfrak{D} = \{(\Theta, y')\}$ using the traditional SE kernel. The hyperparameters of the SE kernel used in $\mathcal{GP}_H$ is again tuned by maximising the log-likelihood mentioned in Eq. (33). The signal variance and the characteristic lengthscale of this SE kernel are tuned in the interval $(0, 2]$ and $(0, 1]$, respectively. Next, we use standard GP-UCB acquisition function to iteratively search for the best hyperparameter set $\Theta$ that maximises the model performance $y'$. Therefore, the kernels approximated from such a tuned Hyper-GP has better generalisation performance.

### A.5.2 Synthetic Experiments

In the experiments with synthetic functions, we have compared our results with the following stationary and non-stationary kernels.

- Squared Exponential (SE) kernel: $k_{\mathrm{SE}}(\mathbf{x}_1, \mathbf{x}_2) = \sigma_f^2 \exp\left(\frac{-1}{2\Upsilon^2}\|\mathbf{x}_1 - \mathbf{x}_2\|^2\right)$, where $\sigma_f^2$ and $\Upsilon$ correspond to the signal variance and the lengthscale hyperparameter of the SE kernel.

- Matérn Kernel with $\nu = 3/2$:
  $k_{\mathrm{MAT}}(\mathbf{x}_1, \mathbf{x}_2) = \left(1 + \frac{\sqrt{3}}{\Upsilon}r\right) \exp\left(-\frac{\sqrt{3}}{\Upsilon}r\right)$, where $r = \|\mathbf{x}_1 - \mathbf{x}_2\|$ and $\Upsilon$ corresponds to the lengthscale hyperparameter.

- Multi-Kernel Learning (MKL) as a weighted combination of SE ($k_{\mathrm{SE}}$), Matérn ($k_{\mathrm{MAT}}$) and Linear kernels ($k_{\mathrm{LIN}} = \mathbf{x}_1\mathbf{x}_2^\intercal + \underline{c}_3$, where $\underline{c}_3 \in (0, 1)$):
  $k_{\mathrm{MKL}}(\mathbf{x}_1, \mathbf{x}_2) = w_1\ k_{\mathrm{SE}}(\mathbf{x}_1, \mathbf{x}_2) + w_2\ k_{\mathrm{MAT}}(\mathbf{x}_1, \mathbf{x}_2) + w_3\ k_{\mathrm{LIN}}(\mathbf{x}_1, \mathbf{x}_2)$

The hyperparameters $\sigma_f^2$, $\Upsilon$ and $\mathbf{w}$ mentioned in the aforesaid kernels are tuned by maximising the log marginal likelihood given by Eq. (33). The hyperparameters $\sigma_f^2$ and $\Upsilon$ are tuned in the interval $(0, 2]$ and $(0, 1]$, respectively and the weights $\mathbf{w}$ are tuned in the interval $[0, 1]$.

We have considered the following functions for our synthetic experiments: **(i)** Triangular wave - NZ (Triangular wave function with non-zero mean and a single peak in the held-out test region), **(ii)** Triangular wave - Z (Triangular wave function with zero mean and two peaks in the held-out test region), **(iii)** a mixture of Gaussian distributions (($\mu_1 = 0.166, \sigma_1 = 0.4$), ($\mu_2 = 0.5, \sigma_2 = 0.9$), and ($\mu_3 = 0.833, \sigma_3 = 0.6$)), and **(iv)** SINC function given by $y(x) = \mathrm{sinc}(x + 10) + \mathrm{sinc}(x) + \mathrm{sinc}(x - 10) + \bar{\epsilon}$, where $\bar{\epsilon} \sim \mathcal{N}(0, 0.001)$ and $\mathrm{sinc}(x) = \frac{\sin(\pi x)}{\pi x}$.

The posterior distributions computed for the aforesaid synthetic functions using KFO and other baselines are as shown in Figure 5. The maximum log-likelihood estimates for the posterior distributions obtained over 10 repeated runs are provided in Table 4. The poor performance of the Squared Exponential (SE) kernel in synthetic experiments is better understood when we compare the posterior distributions of Triangular wave - Z function and Triangular wave - NZ function. The experiment results for Triangular wave - Z function using the SE kernel revealed that the GP predicted mean in the held-out test region does not go downwards as in the Triangular wave - NZ function, instead, it tries to fall back to the $y = 0$ line from both the sides of X-axis, which is the prior mean, thereby explaining the poor likelihood estimates. We emphasise our synthetic results in a sense that the kernel optimised using our proposed approach (KFO) was able to find the correct periodicity in the given input space even without any explicit enforcement of the spatial properties.

We have conducted another set of experiments to demonstrate the kernel recovering capabilities of KFO. In this experiment, we try to recover a non-stationary linear kernel. We have considered a synthetic function - linear function $y = 0.5x$, and generated evenly spaced observations in the interval $[-1, 1]$. Then, we fit a ground truth GP ($\mathcal{GP}_{\mathrm{GT}}$) on those observations using a ground truth kernel ($k_{\mathrm{GT}}$) - a linear kernel in this case, and then we tune the hyperparameters of $k_{\mathrm{GT}}$ by maximising the log-likelihood. Now, we use the same set of observations in the GP ($\mathcal{GP}_{\mathrm{KFO}}$) fitted by our KFO algorithm and run the optimisation procedure to find the optimal kernel. It is evident from Figure 6 that the posterior distributions obtained for $\mathcal{GP}_{\mathrm{KFO}}$ is very identical to that of $\mathcal{GP}_{\mathrm{GT}}$. Further, we have computed the predictive log-likelihood ($PL$) over 500 number of points sampled from a much bigger

range of $[-10, 10]$ for both the GPs. The predictive log-likelihood estimates turned out to be very close as (i) $PL_{\text{KFO}}$= 2449.045, (ii) $PL_{GT}$=2045.148, showing that the kernels are also very close.

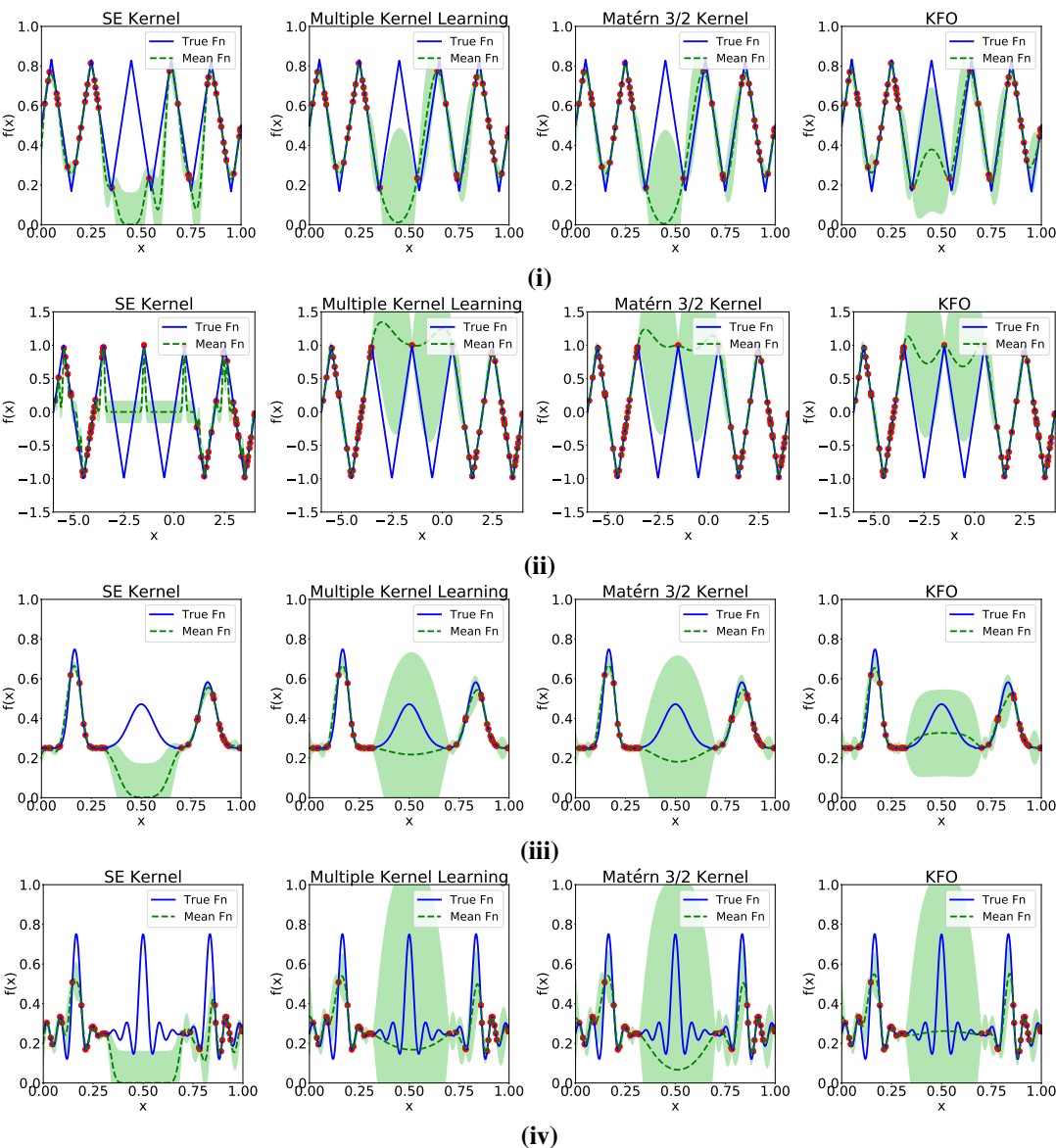

Figure 5: Posterior distributions for (i) Triangular wave - NZ, (ii) Triangular wave - Z, (iii) $G_{\text{mix}}$ , and (iv) SINC functions using KFO and other baselines. The solid blue line shows the true function. The green shaded area covers $2\sigma$ above and below the posterior mean shown by the green dashed line.

Table 4: Log-likelihood computed for the synthetic functions using KFO, SE, Mat3/2 and MKL kernels. Bold indicates the best performance among all the columns. Higher the better.

|  | KFO | SE Kernel | Mat3/2 kernel | Multi Kernel |
|---|---|---|---|---|
| Triangular wave - NZ | **79 ± 1.19** | $-229 \pm 0.4$ | $74.04 \pm 2.08$ | $74.5 \pm 0.7$ |
| Triangular wave - Z | **109.43 ± 0.02** | $-900 \pm 2.24$ | $92.46 \pm 0.82$ | $91.10 \pm 2.08$ |
| $G_{\text{mix}}$ Function | **117.60 ± 2.1** | $-8.58 \pm 1.71$ | $108.36 \pm 1.95$ | $108.89 \pm 1.21$ |
| SINC Function | **82.91 ± 1.74** | $11.62 \pm 1.83$ | $74.1 \pm 1.2$ | $74.98 \pm 0.52$ |

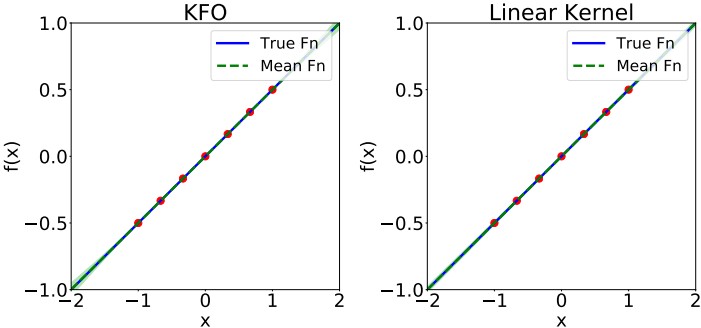

Figure 6: Posterior distributions computed for $\mathcal{GP}_{\text{KFO}}$ and $\mathcal{GP}_{\text{GT}}$.

### A.5.3 Real-world Experiments

**SVM-Classification** The descriptive statistics of the datasets used in the real world experiments are mentioned in Table 5. We demonstrate the effectiveness of our approach using real-world datasets having a diverse set of input characteristics. We use publicly available datasets from UCI repository (Dua and Graff, 2017) in our experiments. In our main paper, we have presented the results comparing our proposed method with the relevant baselines. In addition to that, we demonstrate the superiority of our approach by reporting the best test classification error (the last column marked by † in Table 1 of the main paper) obtained by the state-of-the-art classifiers in the literature (Zhang et al., 2017). To the best of our knowledge, Zhang et al. (2017) is the latest work that surveyed numerous classifiers on UCI datasets. Although we follow the same train/test splits as that of Zhang et al. (2017) for the classification experiments, few classifiers listed in Zhang et al. (2017) are out of the scope of experimental setup we have considered. We just report the results to emphasise on the margin of improvement achieved with our proposed approach.

Table 5: Descriptive statistics of the real-world datasets used.

| Task | Dataset | Features | Instances | Classes |
|---|---|---|---|---|
| Classification | WDBC | 30 | 569 | 3 |
| | Ionosphere | 34 | 351 | 2 |
| | Sonar | 60 | 208 | 2 |
| | Glass | 9 | 214 | 7 |
| | Heart | 13 | 303 | 2 |
| | Seeds | 7 | 210 | 3 |
| | Credit | 24 | 1000 | 2 |
| | Hayes-roth | 5 | 1320 | 3 |
| | Biodeg | 41 | 1056 | 3 |
| | Wine | 13 | 178 | 3 |
| | Ecoli | 8 | 314 | 8 |
| | Car | 6 | 1728 | 4 |
| | Contraceptive | 9 | 1473 | 3 |
| | Phoneme | 5 | 5404 | 2 |
| Regression | Fertility | 9 | 100 | - |
| | Yacht | 7 | 308 | - |
| | Slump | 10 | 103 | - |
| | Boston | 13 | 506 | - |
| | Auto | 7 | 398 | - |
| | Airfoil | 5 | 1503 | - |

We use $C-$SVM in conjunction with KFO to minimise the test classification error. We perform $10-$fold cross-validation on the training data set containing $80\%$ of the total instances and tune

the cost parameter ($C$) of the SVM in the exponent space (base 10) of $[-3, 3]$. In addition to the SVM classification results provided in the main paper, we further evaluate the performance of our proposed approach using the additional baselines described here. We construct a new SVM classifier (KFO-MKL) with its kernel constructed as a weighted combination of KFO tuned kernel and standard parametric kernels to include exogenous kernels in KFO framework *i.e.,* the kernel $k_{\text{KFO-MKL}}$ used in the $C-$SVM classifier is constructed as $k_{\text{KFO-MKL}} = \tilde{w}_1 \; k_{\text{KFO}} + \tilde{w}_2 \; k_{\text{SE}} + \tilde{w}_3 \; k_{\text{MAT}} + \tilde{w}_4 \; k_{\text{LIN}}$. The weights $\tilde{\mathbf{w}}$ are tuned in the interval $[0, 1]$. Furthermore, we compare with $C-$parameterised Linear SVM adhering to the definitions of the hyperkernel optimisation problem using the results mentioned in Ong and Smola (2003).

To show the robustness of our proposed approach, we also provide SVM classification (KFO$_{\alpha-\text{CLIP}}$) results obtained when we clip the values of $\boldsymbol{\alpha}$ in Eq. (12) *i.e.,* $\boldsymbol{\alpha} = [(\alpha_i)_+]$ as an alternative to spectrum clip transformations performed at Gram matrix levels to ensure positive definiteness (see discussion in Sec. 3.2 of the main paper). The classification error percentage obtained for the test set using the aforementioned classifiers averaged over 10 random repeated runs are shown in Table 6.

Table 6: Additional SVM classification results for the real-world datasets using KFO and other baselines, with the test set consisting of 20% of the total instances. Each cell signifies the mean test classification error and the standard deviation computed over 10 runs with random initialisations. Lower the better. Bold indicates the best performance among all the columns.

| Dataset | KFO-MKL | KFO$_{\alpha-\text{CLIP}}$ | $C-$SVM |
|---|---|---|---|
| Ionosphere | **4.28 $\pm$ 0.79** | 5.84 $\pm$ 1.66 | 6.61 $\pm$ 1.82 |
| Glass | 8.65 $\pm$ 1.81 | 13.29 $\pm$ 1.19 | **6.0 $\pm$ 2.4** |
| Sonar | **6.9 $\pm$ 1.66** | 10.04 $\pm$ 2.37 | 14.8 $\pm$ 3.7 |
| Heart | **11.04 $\pm$ 0.8** | 14.13 $\pm$ 1.94 | 19.7 $\pm$ 1.2 |
| Wine | **0** | **0** | **0** |
| Credit | **32.39 $\pm$ 4.6** | 35.6 $\pm$ 2.77 | 35.48 $\pm$ 2.2 |
| Biodeg | **13.81 $\pm$ 0.49** | 19.11 $\pm$ 1.63 | 24.53 $\pm$ 0.8 |
| Hayes-Roth | **17.14 $\pm$ 0.85** | 17.80 $\pm$ 0.79 | 21.43 $\pm$ 2.8 |
| WDBC | 1.96 $\pm$ 0.92 | **1.85 $\pm$ 1.05** | 3.3 $\pm$ 1.2 |
| Contraceptive | **27.63 $\pm$ 4.5** | 38.35 $\pm$ 7.81 | 45.95 $\pm$ 6.2 |
| Car | **0** | **0** | 8.09 $\pm$ 1.27 |
| Phoneme | 29.63 $\pm$ 2.9 | 29.57 $\pm$ 3.86 | **21.52 $\pm$ 1.6** |
| Ecoli | **1.8 $\pm$ 0.9** | 2.05 $\pm$ 0.01 | 15.15 $\pm$ 2.64 |
| Seeds | **1.56 $\pm$ 0.4** | 2.31 $\pm$ 0.78 | 10.61 $\pm$ 2.4 |

**GP-Regression** In the main paper, we have provided an Eigen Value Decomposition (EVD) based approach to transform the Gram matrix by applying a linear transformation at the Gram matrix level to ensure positive definiteness. Though this approach works for SVMs, it may get inadequate for GPs that requires positive definite covariances. Ayhan and Chu (2012) have discussed in detail the issues of using indefinite kernels in GPs. GPs require the calculation of predictive mean $\mu(\cdot)$ and variance $\sigma^2(\cdot)$ for the test samples. It is almost surely impossible to consistently transform the kernel matrix to ensure a positive predictive variance. Therefore, we need ways to enforce positive definiteness before we compute predictive variances. Thus, as an alternative way to ensure positive definiteness, we clip $\boldsymbol{\alpha} = [(\alpha_i)_+]$ mentioned in Eq. (4) of the main paper. The fundamental reason to choose spectrum clipping $\boldsymbol{\alpha} = [(\alpha_i)_+]$ instead of spectrum flipping $\boldsymbol{\alpha} = [|\alpha_i|]$ is that the magnitude of change observed in the transformed matrix is smaller for spectrum clipping. The negative log-likelihood estimates (along with its standard deviation) obtained for GP regression using both clipping (KFO$_{\alpha-\text{CLIP}}$) and flipping (KFO$_{\alpha-\text{FLIP}}$) are listed in Table 7. It is clear from the results that clipping is always better than flipping and in some cases substantially so.

The code base and the relevant modules used in our experiments are available at `https://github.com/mailtoarunkumarav/KernelFunctionalOptimisation`. We have also provided the computing resource requirements of our proposed method.

Table 7: GP Regression results for the real-world datasets using $\text{KFO}_{\alpha-\text{FLIP}}$ and $\text{KFO}_{\alpha-\text{CLIP}}$. Each cell signifies the mean negative log-likelihood and the standard deviation computed over 10 random runs. Lower the better. Bold indicates the best performance among all the columns.

| Dataset | Fertility | Yacht | Concrete Slump | Boston | Auto |
|---|---|---|---|---|---|
| $\text{KFO}_{\alpha-\text{FLIP}}$ | $5.28 \pm 2.7$ | $-13.1 \pm 5.4$ | $-2.97 \pm 1.48$ | $-16.6 \pm 7.03$ | $-8.2 \pm 3.6$ |
| $\text{KFO}_{\alpha-\text{CLIP}}$ | $\mathbf{5.15 \pm 2.9}$ | $\mathbf{-34.6 \pm 1.6}$ | $\mathbf{-3.01 \pm 1.7}$ | $\mathbf{-24.7 \pm 4.2}$ | $\mathbf{-8.7 \pm 1.2}$ |