# OpenReview forum: "Kernel Functional Optimisation"
_NeurIPS.cc/2021/Conference — NeurIPS 2021 Poster_

### Official Review · Reviewer_1xJM · 2021-07-14

**Rating:** 8
**Confidence:** 4

**Summary:**

This article proposes to use the hyperkernel formalism coupled with Gaussian Processes to derive the ‘best’ kernel for regression. The latter kernel depends mostly on the observations rather than on a pre-selected kernel family. The proposed procedure relies on several stages of Bayesian optimization as well as a clipping of the coefficients to avoid dealing with (indefinite) Krein kernels. The performance is presented on both synthetic and benchmark datasets, to assess visually the fit of the confidence intervals as well as the numerical performance.

**Main Review:**

I have found the article to be clearly written, with impressive numerical benchmarks, and a nice balance between exposition of the theory, of the algorithms, of the theoretical and numerical results. Though I did not go through the proofs in the appendix, the latter nicely complements the main body of the article.

Major comments:
-	Clipping: Since for indefinite kernels the strong topology is given by $k_++k_-$, have the authors tried flipping the coefficients ($\alpha_i\rightarrow |\alpha_i |$) instead of clipping them? In the synthetic/numerical experiments, can the authors show the number/magnitude of the coefficients that where clipped? I expect that, since kernels are understood in GPs as generating the covariance matrices, few negative coefficients should appear when fitting the synthetic curves. Could you remind the reader why negative definite covariance matrices cannot be considered in GPs as to explain why a clipping has to be introduced? It seems to me that in ``Learning the Kernel with Hyperkernels`` the discussion after Lemma 7 rules indefinite kernels out only because the authors have decided to focus on RKHSs rather than RKKSs. I thus imagine that the reasons for clipping differ between the kernel and GP communities.
-	Synthetic experiments: In Figure 2 and in the appendix, all the functions appear to have a non-zero mean. In kernel ridge regression (KRR), this would make the task much more difficult for the RBF/SE/Gaussian kernel and I am wondering if this may explain the failure of the SE kernel with this characteristic downward form sticking to zero in the area without observations. Can the authors repeat the synthetic experiments when normalizing the curves to have zero mean? While the choice of a linear kernel is classical, for these types of signals, it would be more adapted if there were some linear trends. Here it just seems to skew the confidence intervals. I believe that if the x-axis was centered to zero, then the weight assigned to the linear kernel would be close to null.
Overall, the Matern 3/2 kernel seems to produce confidence intervals very much alike the ones of the KFO. This begs the question of plotting the kernel of the KFO (for instance by showing $y\mapsto k(x,y)$ for several $x$). Does it recover some translation-invariant features of the data? All the experiments seem to focus exclusively on regression, would the KFO procedure be able to recover the covariance/kernel of a dataset that would be generated with zero mean and covariance some given kernel $k(x,y)$? Can an experiment be done in such a direction?

More generally, since $\lambda$ was managed to be chosen in $[0,1]$, would there be techniques to sample the kernels exclusively in the cone of positive definite kernels? As said above, I liked the article and am willing to upgrade my mark depending on the authors' answers.

Minor comments:

7 I would suggest citing the Ong et al. 2005 paper directly in the abstract since it serves as the main source for the procedure

110 Hilbert with uppercase

119  ``Learning the Kernel with Hyperkernels`` (Lemma 7) could be quoted with profit to justify for the finite decomposition. Or do the authors only intend to give examples of the type of indefinite kernels contained in $H_\kappa$? I very much appreciated the discussion of paragraph 109-120 which denotes a good knowledge of the existing literature.

207 this should be a $\succcurlyeq$

209-214 It is not clear to me why EVD was presented on 196-208 if not used (this contradicts 194 by the way). If the techniques of 196-208 are just quoted as other possibilities that are dismissed, then it should be stated more clearly.

225 Euclidean with uppercase

240 Formulation of Definition 3 is convoluted. I would suggest “if there exists” rather than “such that there exists” and to write $f:H_\kappa \rightarrow H_\kappa$ to make explicit input and output spaces (same for Theorem 2).

241 Why underline the kernels? They have the same interpretation as the ones of Eq. (5).

244 The kernel $k$ is a kernel over the hyper-RKHS, using $k$ here can be confusing. Maybe change it to $\Bbbk$ or any preferred notation?

319 In Table 1, what is $\dagger$ referring to?


**Time Spent Reviewing:**

Three hours

---

> ### Author Response · Authors · 2021-08-10
> **Reply to reviewer 1xJM**
>
> We appreciate our reviewer's patience, time and effort to provide us detailed feedback and raise few interesting points. Please see
> below our replies to the key points raised.
>
> * _"Have the authors tried flipping the coefficients ($\alpha_i\rightarrow|\alpha_i|$) instead of clipping them?"_
>
>     Yes, we did but we had not included these in the current version to avoid potential confusion. The fundamental reason to choose spectrum clipping $\boldsymbol{\alpha}=[{(\alpha_i)}_+]$ instead of spectrum flipping $\boldsymbol{\alpha}=[|\alpha_i|]$ is that the magnitude of change observed in the transformed matrix is smaller for spectrum clipping. The negative log-likelihood estimates (along with its standard deviation) for GP Regression using both clipping and flipping are listed in Table 1. It is clear from the results
> that clipping is always better than flipping and in some cases substantially so.
>
> **Table 1**: GP Regression results in terms of negative log-likelihood and its standard deviation (lower the better).
>
>       |    Dataset    |   Fertility  |     Yacht    |      Slump   |    Boston    |      Auto     |
>       |-------------------------------------------------------------------------------------------|
>       |    KFO_FLIP   |  5.28 ± 2.74 | −13.15 ± 5.4 | −2.97 ± 1.48 | −16.6 ± 7.03 |  −8.21 ± 3.6  |
>       |    KFO_CLIP   |  5.15 ± 2.95 | −34.66 ± 1.6 | −3.01 ± 1.7  | −24.7 ± 4.2  |  −8.78 ± 1.2  |
>       |-------------------------------------------------------------------------------------------|
>
>
> * _"In synthetic experiments can the authors show the number/magnitude of the coefficients that where clipped?"_
>
>     In one synthetic experiment with $N^{2}=16$, we have observed $8$ negative values in $\boldsymbol{\alpha}$ that were eventually removed by clipping. In another similar experimental setup with $N^{2}=81$, we have observed $43$ negative values in $\boldsymbol{\alpha}$ before clipping.
>
>
> * _"Why negative definite covariance matrices cannot be considered in GPs as to explain why a clipping has to be introduced?"_
>
>     For GPs, there is a strong requirement that the covariance matrix is positive definite as it needs to generate positive definite covariances. Ayhan _et al._ (2012) have discussed in detail the issues of using indefinite kernels in GPs. Therefore, we need ways to enforce positive definiteness before we compute predictive variances. In contrast, for SVMs it is possible to use indefinite (Kreĭn) kernels (Alabdulmohsin _et al._ 2015).
>
>
> **Reviewer suggested additional experiments:**
>
> We have conducted few new experiments, but unfortunately we will not be able to show any graph in the current NeurIPS review system. However, we will try to convey the results in the best possible way.
>
> * _"Can the authors repeat the synthetic experiments when normalizing the curves to have zero mean?"_
>
>     As suggested, we have repeated our synthetic experiment to transform our objective function to have zero mean. The new experimental results of our Squared Exponential (SE) kernel revealed that in the held-out test region the GP predicted mean does not go downwards as in the previous case, instead, it tries to fall back to the $y=0$ line from both the sides of X-axis, which is the prior mean. The maximum log-likelihood obtained for all the kernels considered in this experiment is listed in Table 2. The SE kernel still performed
> poorly compared to other kernels.
>
>
> **Table 2**: The maximum log-likelihood (along with its standard deviation) for the synthetic function (higher the better).
>
>       |        Kernel      |       SE     |       MKL     |       MAT     |        KFO      |
>       |-------------------------------------------------------------------------------------|
>       |    Log-likelihood  |  −900 ± 2.24 | 92.468 ± 0.82 | 91.108 ± 0.54 | 109.431 ± 0.026 |
>       |-------------------------------------------------------------------------------------|
>
>
> * _"If the x-axis was centered to zero, then the weight assigned to the linear kernel will be close to null?"_
>
>     We have conducted this new experiment by modifying our objective function to vary between ($-X_{\text{max}}$, $X_{\text{max}}$) such that the objective function is centered on the Origin ($0,0$). The skewness induced in the confidence intervals (an increasing trend in the $\pm2\sigma$) of our previous Multiple Kernel Learning (MKL) posteriors no longer appear in the new experimental results. We further observed that out of all the kernels chosen in MKL, the least weight was assigned to the Linear kernel ($k_{\text{LIN}}$) _i.e.,_ $w_{3}=0.114$ compared to the other weights of $[w_{1,}w_{2}]=[0.478,0.961]$.
>
>
> Both the above-mentioned experiments were merged into one single experiment to measure the maximum log-likelihood for all the kernels considered. The corresponding results are listed in Table 2.
>
>
> * _"Plotting and comparing the optimised Matérn 3/2 kernel and KFO tuned kernel. Can KFO recover translation-invariant data?'"_
>
>     In this experiment, we aim at understanding the characteristics of the kernel tuned with Kernel Functional Optimisation (KFO) procedure $k_{KFO}$ and Matérn
> kernel ($k_{MAT}$) by plotting both the kernels. The results looked visually similar ($3D$ plots). To obtain a numerical comparison
> we have sampled a set of points for the kernel inputs $\mathbf{x_1}$ and $\mathbf{x_{2}}$ and computed their $L_{\infty}$ norm. Based
> on $500$ number of points we estimate the $L_{\infty}$ to be $0.185$, which is rather small. Furthermore, the visual ($3D$ plot) of the
> $k_{\text{KFO}}$ kernel revealed the translation-invariant features (apparent translation invariance along the line $x=y$, reflection
> symmetry on the line $x=-y$).
>
>
> * _"Would KFO procedure be able to recover the kernel of a dataset that would be generated with zero mean and some kernel $k(x,x')$?"_
>
>     We have conducted another set of experiments to demonstrate the kernel recovery by KFO. In this experiment, we try to recover a non-stationary linear kernel. We have considered a synthetic function - linear function $y=0.5x$, and generated $20$ evenly spaced observations in the interval $[-1,1]$. Then, we fit a ground truth GP ($GP_{GT}$) on those observations using a ground truth kernel ($k_{GT}$) - a linear kernel in this case, and then we tune the hyperparameters of $k_{GT}$ by maximising the log-likelihood. Now, we use the same set of observations in the GP ($GP_{KFO}$) fitted by our KFO algorithm and run the optimisation procedure to find the optimal kernel. We visually confirmed that the posterior distribution of $GP_{KFO}$ is identical to that of $GP_{GT}$. Further, we have computed the predictive log-likelihood ($PL$) over $500$ number of points sampled from a much bigger range of $[-10,10]$ for both the GPs and they turned out to be very close as (i) $PL_{KFO}$= $2449.045$ (ii) $PL_{GT}$= $2045.148$, showing that the kernels are also very close.
>
>
> * _"Would there be techniques to sample the kernels exclusively in the cone of positive definite kernels?"_
>
>     This is an interesting question that we have pondered before. One possible way to ensure that the kernel remains positive is by constructing the kernel samples $\mathbf{k}$ (Eq. (5) of the main paper) by sampling strictly positive $\boldsymbol{\beta}^{(\cdot)}$. Then, we apply a linear transformation on $\mathbf{k}$ to ensure its positive definiteness. Further, we restrict the kernel weights to be strictly positive (Eq. (4) of the main paper). We believe that such constraints result in a highly restricted search space, not covering the full cone of positive definite kernels. The optimisation on such an overly restricted search space results in sub-optimal solutions, which may not be acceptable. Unfortunately, we are unable to construct any different method that can restrict sampling in the cone of positive definite kernels.
>
> **_Minor comments addressed:_**
> * _Line 119 : "do the authors only intend to give examples of the type of indefinite kernels contained in_ $\mathcal{H}_{\kappa}$ _?"_
>
>
>     We initially intended to discuss only the decomposition with respect to indefinite kernels, but it is a decent idea to quote Lemma 7 of
> Ong _et al._ (2005) to complement to what we are trying to state.
>
>
> * _Line 209_ : In _Line 196-208_, we have provided an approach to transform the kernel matrix by applying a linear transformation
> at the Gram matrix level. Though this approach works for SVMs, it may get inadequate for GPs. GPs require the calculation of predictive
> mean $\mu(\cdot)$ and variance $\sigma^{2}(\cdot)$ for the test samples. It is almost surely impossible to consistently transform the kernel matrix to ensure a positive predictive variance. Therefore, as an alternative way to ensure positive definiteness in GPs, we clip $\boldsymbol{\alpha}=[{(\alpha_i)}_{+}]$ mentioned in Eq. (4) of the main paper.
>
> * _Line 319_: In Table 1 of the main paper, the column marked by $\dagger$ corresponds to the accuracy results obtained by state-of-the-art classifiers listed in Zhang et al. (2017). We have just reported the results from other classifiers present in the literature to emphasise the margin of improvement achieved with our proposed approach. The details are discussed in _Line 286-292_.
>
>
> **References**
>
> * C. S. Ong, A. J. Smola, and R. C. Williamson. Learning the kernel with hyperkernels. Journal of Machine Learning Research, 6(Jul):1043--1071, 2005.
>
> * Ayhan, M. Seckin, and C. H. Chu. Towards indefinite gaussian processes. Technical report, University of Louisiana at Lafayette, 2012.
>
> * I. Alabdulmohsin, X. Gao, and X. Zhang. Support vector machines with indefinite kernels. In Asian Conference on Machine Learning, pp. 32-47. PMLR, 2015.
>
> * C. Zhang, C. Liu, X. Zhang, and G. Almpanidis. An up-to-date comparison of state-of-the-art classification algorithms. Expert Systems with Applications, 82:128--150, 2017.

---

> > ### Comment · Reviewer_1xJM · 2021-08-23
> > **Answer**
> >
> > I thank the authors for their in-depth reply. I would highly encourage them to incorporate all the comments above either in the main text or in the supplementary. Especially, since Krein kernels are a more confidential topic in the kernel community, stressing the relevance for kernels and GPs of using nonnegative covariances can clarify the ways to handle indefinite kernels in practice.

---

> > > ### Author Response · Authors · 2021-08-26
> > > **Thanks for the acknowledgement**
> > >
> > > We thank the reviewer for reading and acknowledging our rebuttal. We will incorporate the reviewer suggestions and comments into the paper if it gets accepted.

---

### Official Review · Reviewer_ngk1 · 2021-07-16

**Rating:** 8
**Confidence:** 4

**Summary:**

Paper presents a novel framework for Bayesian learning of kernels using hyperkernels. It is able to address a broader set of kernels that are stationary and non-stationary, learn them efficiently and show state-of-the-art results on synthetic and real-world datasets.

**Ethical Concerns:**

No specific ethical concerns.

**Limitations And Societal Impact:**

Paper addresses technical limitations well but does not address effect of model/data bias and explainability of learned models.

**Main Review:**

A key contribution of the paper is of the ability to address a broader set of kernels that are non-stationary by using indefinite kernels later approximating positive-definite projection.  This allows the kernel regression approximation to deal with sharp changes in function value as shown in synthetic data.

Paper also also provides sound theoretical justification in terms of regret convergence of  of proposed algorithm as parameters of the effective dimension.

Projection of data to to S subspaces provides a scalable approach to deal with larger datasets but parameter selection is not clear (n, T, S) seem to best selected for UCI datasets but selection strategy is not clear. How would parameter selection work for datasets like ImageNet and so on.

Evaluation is shown on UCI dataset which are real-word but small and not clear if it the proposed technique also works for large scale100s of category's classification.

Table 1 and Table 2 show that KFO does well for almost all but a few datasets. For classification Credit, Biodeg and Phoneme other classifier do well and for regression Fertility dataset ARD Matern does better. Is there a explanation for such a difference ? Are there any dataset specific peculiarities when KFO does not do well?






**Time Spent Reviewing:**

4

---

> ### Author Response · Authors · 2021-08-10
> **Reply to reviewer ngk1**
>
> We thank our reviewer's time and effort invested in understanding the paper to raise some interesting points. Please see
> below our replies to the key points raised.
>
>
> * _"Parameter selection is not clear (n, T, S)"_
>    * $n$ is the dimensionality of the underlying classification or regression problem and is classification/regression problem-specific.
>
>
>    * $T$, $S$ are the parameters of our Kernel Functional Optimisation (KFO) method, where $S$ is the number of subspace searches and $T$ is the number of iterations in each of the subspaces.
>
>     In this work, we have considered a fixed computational budget ($S\times T$) for the optimisation and set $S=5$ and $T=20$. If we are not concerned with the computational budget, we can set up stopping criteria (following Kirschner _et al._ (2019)) based on the quality of the solution achieved _i.e.,_ the terminating condition is set with a measure such as simple regret or cumulative regret to keep track of the optimisation performance. We have discussed the convergence guarantees of our approach for both the criteria (fixed and variable $T$) in _Theoretical Analysis_ section (section _4.3_)
>
>
> * _"How would parameter selection work for datasets like ImageNet?"_
>
>     The main bottleneck of our proposed approach is the computation of the covariance matrix $\boldsymbol{\kappa}\in\mathbb{R}^{N_{g}\times N_{g}}$, where $N_{g}$ corresponds to the number of points in the grid (Note: for notational convenience we use $N_g$ for the grid size). The computational complexity of performing Principal Component Analysis
> (PCA) to identify $N'\ll N_{g}^{2}$ principal components of $\boldsymbol{\kappa}$ is $\mathcal{O}(N'N_{g}^{2})$. Following our same recipe on ImageNet dataset with $n=1024$, we can choose the grid size $N_{g}=10\times1024=10240$ to compute the covariance matrix $\boldsymbol{\kappa}$, which is feasible on HPC infrastructure, and then reduce the complexity to a more reasonable $N'\ll N_{g}^{2}$ prior to entering the main optimisation loop. The main Bayesian optimisation loop in our proposed approach is not tightly coupled with $n$ and remains unaffected by the computation and PCA decomposition of the covariance matrices. Thus we may still be able to run KFO on this dataset, although we may be approaching the limit of feasibility. However, on a more practical note, kernel algorithms excel in small to middle-sized problems and KFO should be easily applicable in such scenarios.
>
>
> * _"If the proposed technique also works for large scale100s of category's classification"_
>
>     The number of categories or the target classes in the given problem does not play a role in determining the optimisation performance.
> Rather, this is determined by the complexity of the kernel search space, which may or may not be coupled to such factors.
>
>
> * _Dataset specific peculiarities of KFO_
>
>     We ideally expect our proposed method KFO to at least achieve the performance of any standard parametric kernel. However, it is observed from our empirical results that sometimes our approach is not able to find the optimum kernel, the possible reason could be the insufficient computational budget allocated or the numerical approximations.
>
>
> * _Model/data bias_
>
>     Since we use a universal hyperkernel for constructing the space of kernel functionals on which we perform optimisation, there is no bias
> induced into our models. Data bias and explainability are outside the scope of our paper.
>
>
> **References**
>
> * J. Kirschner, M. Mutny, N. Hiller, R. Ischebeck, and A. Krause. Adaptive and safe Bayesian optimisation in high dimensions via one-dimensional subspaces. arXiv preprint arXiv:1902.03229, 2019.

---

### Official Review · Reviewer_Fbiv · 2021-07-18

**Rating:** 5
**Confidence:** 4

**Summary:**

This paper introduces a novel approach which constructs the best fitting kernel function with adaptive tuned parameters. This kernel function is obtained by the linear combination of multiple kernels sampled from a prior Gaussian Process. The experimental results of two different kernel-based algorithms show the superiority of the proposed method.

**Limitations And Societal Impact:**

However, there are still some issues should be addressed as follows:
1. In Line 163, \beta is a vector and should be in boldface. Moreover, the author should use upper case to represent the test set.
2. This paper involves a lot of basic knowledge. The author should give more detailed introduction.
3. What does GP-UCB mean in the Line 76? To increase the readability, the author should give some necessary descriptions in the main paper.
4. A PCA algorithm is performed on a N^2 * N^2 matrix to avoid the computational burden as described in Line 157-159. However, the complexity of PCA is cubic of the dimension of a squared matrix. Please explain how PCA can reduce the complexity.

**Main Review:**

This paper introduces a novel approach which constructs the best fitting kernel function with adaptive tuned parameters. This kernel function is obtained by the linear combination of multiple kernels sampled from a prior Gaussian Process. The experimental results of two different kernel-based algorithms show the superiority of the proposed method.

**Time Spent Reviewing:**

15

---

> ### Author Response · Authors · 2021-08-10
> **Reply to Reviewer Fbiv**
>
> We appreciate your effort to raise important points and provide valuable suggestions. Please see below our replies to the key points raised.
> * _"This paper involves a lot of basic knowledge. The author should give more detailed introduction"_
>
>     In the current version of our paper, we tried our best to provide all the necessary background given the space constraints. But, we
> will try to rewrite few sections to create some free space and incorporate the given feedback.
>
>
> * _"GP-UCB? To increase the readability, the author should give some necessary descriptions in the main paper"_
>
>    In _Line 76_, the acronym GP-UCB stands for Gaussian Process - Upper Confidence Bound, one of the popular acquisition functions
> commonly used in the context of Bayesian optimisation. We have briefly discussed in _Background_ section, as well as in the supplementary material (section _A.1.1_). A detailed explanation of GP-UCB and its theoretical guarantees are provided in Srinivas _et
> al._ (2012) and Brochu _et al._ (2010).
>
>
> * _"Explain how PCA can reduce the complexity?"_
>
>     In _Line 157-159_, we perform Principal Component Analysis (PCA) to avoid the computational burden resulting from the large covariance matrix $\mathbf{\boldsymbol{\kappa}}\in\mathbb{R}^{{N_g}\times{N_g}}$ for the given grid size $N_{g}$ (Note: for notational convenience we use $N_g$ for the grid size). Here, we do not perform a full PCA, rather we choose only top $N'$ principal components ($N'\ll N_{g}$). The computational complexity of finding top $N'$ principal components is $\mathcal{O}(N'N_{g}^{2})$, which is much lower than $\mathcal{O}(N_{g}^{3})$. Moreover, we perform PCA only once, prior to entering the outer and inner optimisation loops. Thus we incur a cost on startup but are rewarded with significant computational savings in the main optimisation loop where the computational burden is proportional to $N'$ rather than $N_{g}^{2}$.
>
> **References**
> * E. Brochu, V. M. Cora, and N. De Freitas. A tutorial on Bayesian optimisation of expensive cost functions, with application to active user modeling and hierarchical reinforcement learning. arXiv preprint arXiv:1012.2599, 2010.
>
> * N. Srinivas, A. Krause, S. M. Kakade, and M. W. Seeger. Information-theoretic regret bounds for Gaussian process optimisation in the bandit setting. IEEE Transactions on Information Theory, 58 (5):3250--3265, 2012.

---

### Official Review · Reviewer_Bqt6 · 2021-07-18

**Rating:** 7
**Confidence:** 3

**Summary:**

The paper proposes a zero-order optimization method where the optimized variable is a kernel function in Hyper-RKHS induced by a selected hyper-kernel. The method is an instance of the Bayesian Optimization method LINEBO [Kirschner 2019]. The contribution is in the adaptation of the LINEBO algorithm for efficient optimization w.r.t. positive definite kernel functions. The algorithm is applied to the optimization of kernel functions for C-SVM and GP regression. Experiments show significant improvement over existing methods.

**Limitations And Societal Impact:**

yes

**Main Review:**

Originality. Using the Bayesian optimization for kernel selection has been previously proposed in

Malkomes et al. Bayesian optimization for automated model selection. NIPS 2016.

There are similarities between the proposed method and the mentioned paper. For example, both use the GP to model the expensive objective function, the performance measure, and both optimize the objective via Bayesian optimization. There are also differences. Most notably, in the mentioned paper the kernel space is defined in terms of base kernels and a grammar to combine them, instead of using the Hyper-RKHS like in the paper under review. More detailed comparison of the two papers is work of the authors, who unfortunately do not reference it.

Quality. The paper is technically sound. The authors provide a convergence analysis of the proposed method.

Clarity. The paper is clearly written.

One unclear point that needs more discussion is the way used to select $N^2$ points for representing the hyper-kernel. The authors state that the points are constructed such that they represent the kernel sufficiently well (lines 142-143), however, it is not clear how.

Significance. The empirical evaluation shows promising results. However, there are are points that need to be clarified. The paper does not provide a clear discussion of the computational/memory demands of the method and its efficacy w.r.t. size of the data (the number of examples and the feature space dimension) and the hyper-parameters like the number of approximation points $N^2$. There should be at least rough information about the computational time needed to perform the experiments on the data used.


**Time Spent Reviewing:**

5

---

> ### Author Response · Authors · 2021-08-10
> **Reply to reviewer Bqt6**
>
> We appreciate your time and effort to provide a thoughtful review. Please see below our replies to the key points raised.
> * _Comparison with "Bayesian optimization for automated model selection"_
>
>    We thank you for the reference of Malkomes _et al._ (2016). We will add this in our related work section. However, as pointed out by you, our work is fundamentally different - our method can obtain the best non-parametric kernel, whereas the design space for Malkomes
> _et al._ (2016) is limited in the compositional space of parametric forms. In short, Kernel Functional Optimisation (KFO) designed kernel
> should supersede any kernels found by them.
>
>
> * _Selection of $N^{2}$ points for representing the hyperkernel_
>
>    The number of points $N^{2}$ in the grid $\mathcal{G}$ is an important parameter. A very fine resolution grid ensures that we can capture small-scale patterns in the kernel. However, a large grid size comes with large computational cost. As a trade-off we set the grid size _i.e.,_ $N^{2}\geq10\times n$, and we find that this works in practice (specifically, (a) it outperforms the baselines in majority
> of the datasets, and (b) we observe that increasing $N^{2}$ further does not give significant improvements in performance). Also, to improve clarity we will replace the notation $N^{2}$ by $N_{g}$ in the updated version.
>
>
> * _Discussion of the computational/memory demands of the method and its efficacy w.r.t. size of the data_
>
>     The computational complexity of our approach is in the order of $\mathcal{O}(STN_{g}^{3})$, where $S$ is the number of subspace searches, $T$ is the number of iterations in each subspace and $N_{g}$ is the number of points in the grid, not including the complexity of the downstream class (as it would be different for different kernel machines). The memory complexity is in the order of $\mathcal{O}(N_{g}^{2})$. The percentage of time spent optimising (searching) the kernel ($4^{th}$ column in the table) and the percentage of time spent evaluating the kernel ($5^{th}$ column in the table) by our proposed approach is shown in the table below. We can see that the percentage of time spent in optimising the kernel is no more than $10\\%$ of the whole model fitting time. Hence, the proposed
> KFO does not add much overhead to the model fitting process. We have also provided the total runtime (in seconds) required for an instance of KFO tuned SVM to complete $S\times T$ iterations, where $S=T=5$. The total runtime also includes the runtime required for generating $4$ random observations in each subspace. The wall clock times for our experiments are measured on a server with Intel(R) Xeon(R) processor, having 16 GB of RAM.
>
> **Table**: Runtimes measured for SVM tuned with KFO kernel
>
>       | Dataset | Instances | Features |  Search Time(%) |  Evaluation Time(%) |  Total Runtime  |
>       |------------------------------------------------------------------------------------------|
>       | Sonar   |    208    |    60    |      9±0.30     |     90.87±0.69      |   567.85±4.64   |
>       | Heart   |    303    |    13    |    4.7±0.35     |     95.24±0.56      |  1070.03±24.12  |
>       | Glass   |    214    |    9     |   9.35±0.55     | 	90.64±0.44      |   554.73±8.19   |
>       | Credit  |    1000   |    24    |   0.90±0.13     |     98.84±0.36      |  5191.36±42.38  |
>       |------------------------------------------------------------------------------------------|
>
>
>
> **References**
> * G. Malkomes, C. Schaff, and R. Garnett. Bayesian optimisation for automated model selection. JMLR Work. Conf. Proceedings; ICML 2016 AutoML Work., 2016.

---

> > ### Comment · Reviewer_Bqt6 · 2021-08-16
> > **Thanks for the reply**
> >
> > I thank the authors for the replies to the issues raised in my review. All the info provided in the reply should be included into the paper if accepted.

---

> > > ### Author Response · Authors · 2021-08-26
> > > **Thanks for the acknowledgement**
> > >
> > > We thank the reviewer for reading and acknowledging our rebuttal. We will incorporate the reviewer suggestions and comments into the paper if it gets accepted.

---

### Official Review · Reviewer_gTjs · 2021-07-20

**Rating:** 5
**Confidence:** 3

**Summary:**

This work is interested in kernel selection. In contrast with most of the literature they consider non-parametric class of kernels using hyper-kernels. Nevertheless, some previous works have already consider this setting but the authors argue that they had several limitations which motivates a new method. The authors first recall the background on Bayesian optimization and hyper-kernels then explain their method to mix the two while avoiding some computational complexity issues. They also show how to fix a naturally arising problem : the posterior mean of the hyper-GP may be positive. Finally, some theoretical and experimental results are provided.

**Limitations And Societal Impact:**

-

**Main Review:**

The problem of kernel selection with non-parametric class of kernels is a subject of interest which has already been investigated. The limitations of the previous works indeed motivate new methods. It would have been great to have a more in-depth comparison with those works after the presentation of KFO to show that the latter does not suffer the same limitations.

Section 2 and 3 are clear enough.
The weak point of this work is the theoretical analyses of KFO. Indeed, the theorems may be stated in a more precise way : recall the space of each object, highlight the main claim with a centering when possible, recall the definition of each quantity or give a precise reference with equation numbers.
The proofs are not easy to check mainly because they are too wordy where a sequence of equations would be easier to follow. The use of previous results could also be more precise. The paragraph l.206 to l.216 is one example of those two problems.
It would be great to discuss more the results, comparing it to previous works on non-parametric kernel selection. Assumption 2 of Kirschner et al. 2019 should be recalled.

The experimental part seems more interesting.
Overall, this work has a good potential but the theoretical part should be polished.

Typos:
- l.17 parameterise -> parameterize
- l.79 symbol of average regret
- l.241 space of f ? if f(K) is real no need for ||.||_{H_k}.

**Time Spent Reviewing:**

6

---

> ### Author Response · Authors · 2021-08-10
> **Reply to reviewer gTjs**
>
> We appreciate your valuable comments and overall feedback. Please see below our replies to the key points raised.
> * _"In-depth comparison of KFO with related works to show that it does not suffer the same limitations"_
>
>     Ong _et al._ (2005) and Benton _et al._ (2019) are the main work in the space of non-parametric kernel design. We have discussed
> their work and the key limitation in _Line 21-32_ and in contrast how our proposed method overcomes these limitations in _Line
> 33-38_.
>
>
> * _"The weak point of this work is the theoretical analyses of KFO"_
>
>    In _Theoretical Analysis_ section, the paragraph written after the proof of _Lemma 2_ is just a remark following the formal proof. We thought that adding a remark section would help our readers to visualise how we use the existing results (from Kirschner _et
> al._ (2019)) to support and prove our claims. However, based on your feedback we will have a go at rewriting it.
>
> **References**
>
> * C. S. Ong, A. J. Smola, and R. C. Williamson. Learning the kernel with hyperkernels. Journal of 378 Machine Learning Research, 6(Jul):1043--1071, 2005.
>
> * G. Benton, W. J. Maddox, J. Salkey, J. Albinati, and A. G. Wilson. Function-space distributions over kernels. In Advances in Neural Information Processing Systems (NeurIPS 2019), pages 14965--14976, 2019.
>
> * J. Kirschner, M. Mutny, N. Hiller, R. Ischebeck, and A. Krause. Adaptive and safe Bayesian optimisation in high dimensions via one-dimensional subspaces. arXiv preprint arXiv:1902.03229, 2019.

---

> > ### Comment · Reviewer_gTjs · 2021-09-01
> > **Thanks for the reply**
> >
> > Thank you for your response. The other reviews have highlighted several qualities of the paper that make me increase my score from 4 to 5. I encourage the authors to polish the theoretical part if the paper gets accepted.

---

> > > ### Author Response · Authors · 2021-09-02
> > > **Thanks for the acknowledgement**
> > >
> > > We thank the reviewer for acknowledging our rebuttal. We will incorporate the reviewer suggestions into the paper if it gets accepted.

---

### Decision · Program_Chairs · 2021-09-27

**Decision:**

Accept (Poster)

**Comment:**

The paper proposes a zero-order optimization method where the optimized variable is a kernel function in Hyper-RKHS induced by a selected hyper-kernel. The algorithm is applied to the optimization of kernel functions for C-SVM and GP regression. Experiments show significant improvement over existing methods. The reviewers consider the paper technically sound and clearly written with a potential relevant impact on the field. We encourage the authors to improve the clarity of the mathematical part following the discussion that emerged during the review and the rebuttal period.